# TRANSFERRING INDUCTIVE BIASES THROUGH KNOWLEDGE DISTILLATION

## ABSTRACT

Having the right inductive biases can be crucial in many tasks or scenarios where data or computing resources are a limiting factor, or where training data is not perfectly representative of the conditions at test time. However, defining, designing, and efficiently adapting inductive biases is not necessarily straightforward. Inductive biases of a model affect its generalization behavior and influence the solution it converges to from different aspects. In this paper, we investigate the power of knowledge distillation in transferring the effects of inductive biases of a teacher model to a student model, when they have different architectures. We consider different families of models: LSTMs vs. Transformers and CNNs vs. MLPs, in the context of tasks and scenarios with linguistics and vision applications, where having the right inductive biases is critical. We train our models in different setups: no knowledge distillation, self-distillation, and distillation using a teacher with a better inductive bias for the task at hand. We show that in the later setup, compared to no distillation and self-distillation, we can not only improve the performance of the students, but also the solutions they converge become similar to their teachers with respect to a wide range of properties, including different task-specific performance metrics, per sample behavior of the models, representational similarity and how the representational space of the models evolve during training, performance on out-of-distribution datasets, confidence calibration, and finally whether the converged solutions fall within the same basins of attractions.

## 1 INTRODUCTION

Inductive biases are the characteristics of learning algorithms that influence their generalization behavior, independent of data. They are one of the main driving forces to push learning algorithms toward particular solutions (Mitchell, 1980). Having the right inductive biases is especially important for obtaining high performance when data or compute is a limiting factor, or when training data is not perfectly representative of the conditions at test time. Moreover, in the absence of strong inductive biases, a model can be equally attracted to several local minima on the loss surface; and the converged solution can be arbitrarily affected by random variations like the initial state or the order of training examples (Sutskever et al., 2013; McCoy et al., 2020; Dodge et al., 2020).

There are different ways to inject inductive biases into learning algorithms, for instance through architectural choices, the objective function, the curriculum, or the optimization regime. In this paper, we exploit the power of *Knowledge Distillation* (KD) to transfer the effect of inductive biases between neural networks. KD refers to the process of transferring knowledge from a teacher model to a student model, where the logits from the teacher are used to train the student. KD is best known as an effective method for model compression (Buciluǎ et al., 2006; Hinton et al., 2015; Sanh et al., 2019) which allows taking advantage of a huge number of parameters during training while having an efficient smaller model during inference.

The advantage of KD goes beyond model compression and it can be used to combine strengths of different learning algorithms (Kuncoro et al., 2019; 2020). Different algorithms vary in terms of the computational/memory efficiency at training/inference or the inductive biases for learning particular patterns. This makes them better at solving certain problems and worse at others, i.e., there is no "one size fits all" learning algorithm. Hence, it is important to explore the potential of KD for finding better trade-offs.

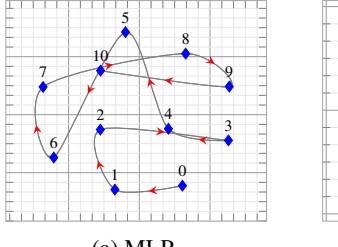 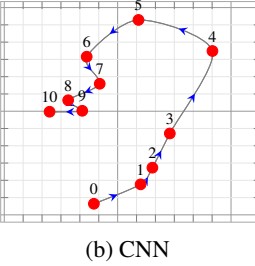 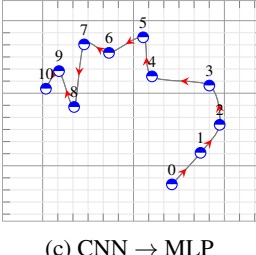

|  (a) MLP  |  (b) CNN  |  (c) CNN → MLP  |

Figure 1: Training paths of different models on the Translated MNIST task. Different points represent the state of the model at different epochs, from the initial state to the convergence. The visualization is based on a 2D projection of the representational similarity of the activations from the penultimate layer for the examples from the validation set, i.e. Translated MNIST (more details in Appendix B).

The question that we ask in this paper is: "*In KD, are the preferences of the teacher that are rooted in its inductive biases, also reflected in its dark knowledge[1], and can they thus be transferred to the student?*". We are interested in cases where the student model can realize functions that are realizable by the teacher, i.e., the student model is efficient with respect to the teacher model (Cohen et al., 2016), while the teacher has a *preference inductive bias* so that the desired solutions are easily *learnable* for the teacher (Seung et al., 1991).

We consider two scenarios where the teacher and the student are neural networks with heterogeneous architectures, hence have different inductive biases. We train the models, both independently and using KD, on tasks for which having the right inductive biases is crucial. In the first test case, we study RNNs vs. Transformers (Vaswani et al., 2017), on the subject-verb agreement prediction task (Linzen et al., 2016). In this task, we use LSTMs (Hochreiter & Schmidhuber, 1997) as the most widely used RNN variant. LSTMs are shown to perform better than vanilla Transformers in this task and their superior performance is attributed to their so-called "recurrent" inductive bias (Tran et al., 2018). First, we identify the sources of the recurrent inductive bias of LSTMs: *sequentiality*, *memory bottleneck*, and *recursion*, and design experiments to show the benefits of each. Then, we show that through distilling knowledge of LSTMs to Transformers, the solutions that the Transformer models learn become more similar to the solution learned by LSTMs.

In the second test case, we study CNNs vs. MLPs, in the context of the MNIST-C (Corrupted MNIST) benchmark (Mu & Gilmer, 2019), which is designed to measure out-of-distribution robustness of models. We train our models on MNIST and evaluate them on the Translated/Scaled MNIST. The particular form of parameter sharing in CNNs combined with the pooling mechanism makes them equivariant to these kinds of transformations (Goodfellow et al., 2016), which leads to better generalization in these scenarios compared to MLPs.

In our experiments and analysis on these two test cases[2], we compare the behavior of different models, from a wide range of perspectives, when trained in different setups including **(1)** when trained without KD, but directly from the data, **(2)** when trained with KD using a teacher with a similar architecture to the student, i.e. self-distillation, and **(3)** when trained with KD using a teacher with a different architecture that has stronger inductive biases that suit the task, compared to the student.

As the first step, in setup (1), i.e., no KD, we demonstrate how inductive biases arising from different architectural choices affect the generalization behavior of the models we study (§2.1 and §3.1). We show that the models with more suitable inductive biases not only have better accuracy but also the solutions they converge to is a better solution in terms of other metrics. We also show that different instances of the model with stronger inductive biases have less variance in terms of all the metrics.

Then, we apply KD to train the models and contrast the behavior of models trained with the setups (2) and (3) with the models trained with setup (1), i.e. with KD vs. without KD. We show that regardless of the properties of the teacher, KD is a powerful technique in which the teacher model drives the student toward a particular set of solutions that is more restricted compared to the set of possible solutions that a student can converge to when it learns directly from data (§2.2, §3.2, and Appendix C).

---

[1] Dark knowledge refers to the information encoded in the output logits of a neural network (Hinton et al., 2015).

[2] The codes for the input pipelines, models, analysis, and the details of the hyper-parameters used in our experiments is available at `https://ANONYMIZED`.

Next, as the main contribution of our paper over previous works that study KD, we contrast the behavior of models trained with setup (3) with the models trained with setups (1) and (2):

- We show the performance of the student models in setup (3) increases, not only on in-distribution test sets (§2.2), but also on out-of-distribution data (§3.2). We demonstrate that this happens when the teacher has the right inductive bias and not necessarily otherwise, i.e., setup (2).

- In setup (3), besides performance, we show that, the solution that a student model converges to shares similar characteristics with the solution of its teacher. For instance in terms of confidence calibration (§2.2 and §3.2), and per sample behaviour of the model (Appendix E).

- We demonstrate that although the student model is merely exposed to the final logits of the teacher, the structure of the latent space of the student model becomes similar to the teacher, i.e. relational similarity of the internal representations from the student and its teacher increases (§2.2 and §3.2).

As an example, in our second test case (MNIST-C), when training an MLP model with KD using a CNN teacher, the student model explores the solution space in ways more similar to its teacher. Figure 1 visualizes and compares the path that an MLP takes during training (Figure 1a), compared to a CNN (Figure 1b). The CNN model explores the surface in a completely different manner than the MLP, while the path of a student MLP distilled from the CNN model as the teacher (Figure1c) is more similar to the CNN.

## 2 DISTILLING LSTMS INTO TRANSFORMERS

LSTMs and Transformers are the basic building blocks of many state-of-the-art models for sequence modeling and natural language processing. Transformers are an expressive class of models that do extremely well on many tasks where the training data is adequate in quantity (Devlin et al., 2019; Keskar et al., 2019; Radford et al., 2019; Brown et al., 2020). Several studies, however, have shown that LSTMs can perform better than Transformers on tasks requiring sensitivity to (linguistic) structure, especially when the data is limited (Tran et al., 2018; Dehghani et al., 2019).

We chose the subject-verb agreement prediction task, introduced by Linzen et al. (2016), as the test case, as it yields a meaningful difference between LSTMs and Transformers (Tran et al., 2018). We compare these two families of models and conduct experiments to emphasize the differences between them when trained independently and through KD.

**Recurrent Inductive Bias.** Among sequence modeling architectures, models with recursion are in particular powerful for natural language processing due to their adequacy to model hierarchical structures (Linzen et al., 2016). The recursion in a model can be implemented in various ways, like in Recurrent Neural Networks (Elman, 1990), Recursive Neural Networks (Socher et al., 2010; Le & Zuidema, 2014) and Universal Transformers (Dehghani et al., 2019; Hao et al., 2019). While theoretically, both recurrent neural networks (RNNs) and Transformers can deal with finite hierarchical structures, empirical results indicate the superiority of RNNs over Transformers (Tran et al., 2018; Dehghani et al., 2019; Hahn, 2020).

In the literature (Sutskever et al., 2013; Dehghani et al., 2019), the inductive bias of RNNs is referred to as the *recurrent inductive bias*. Here, we distinguish between three main sources of this bias:

1. **Sequentiality**: There is an inherent notion of order in the architecture that forces the model to access the next tokens in the input one by one and process them sequentially.
2. **Memory bottleneck**: The model has no direct access to the past tokens and has to compress all the information from the past in a hidden state, which is accessible when processing a new token.
3. **Recursion**: The model recursively applies the same function on the varying input at every step.

Transformers (Vaswani et al., 2017), in contrast, process the input in parallel. Although a weak notion of order is encoded by positional embeddings, no explicit assumption is made in the connectivity structure of the architecture. Moreover, they have a global receptive field and can access all tokens through self-attention. Finally, standard Transformers are not recursive. However, the standard Transformer can be modified to have an architecture with specifications that are similar to RNNs. We provide empirical results to demonstrate the benefits of these different sources of inductive biases of RNNs. For this purpose, we design experiments with variants of Transformers in which we attempt to approximate some of the RNNs' assumptions.

Table 1: Performance (mean±std over 4 trials) of different LSTM and Transformer models trained independently with the LM objective.

| Model | Perplexity ↓ | $\mathcal{D}-$**Accuracy** ↑ | $\mathcal{A}-$**Accuracy** ↑ |
|---|---|---|---|
| **Transformer** | $57.50 \pm 0.1199$ | $0.9417 \pm 0.0017$ | $0.9191 \pm 0.0018$ |
| **Small Transformer** | $\mathbf{55.31} \pm 0.0847$ | $0.9467 \pm 0.0012$ | $0.9261 \pm 0.0020$ |
| **LSTM** | $56.68 \pm 0.0906$ | $\mathbf{0.9510} \pm 0.0012$ | $\mathbf{0.9400} \pm 0.0024$ |
| **Small LSTM** | $58.05 \pm 0.1141$ | $0.9491 \pm 0.0006$ | $0.9366 \pm 0.0015$ |

Table 2: Performance (mean±std over 4 trials) of different LSTM and Transformer models trained independently with the classification objective.

| Model | $\mu-$**Accuracy** ↑ | $\mathcal{D}-$**Accuracy** ↑ | $\mathcal{A}-$**Accuracy** ↑ |
|---|---|---|---|
| **Transformer** | $0.954 \pm 0.0016$ | $0.901 \pm 0.0037$ | $0.717 \pm 0.0244$ |
| **Transformer-seq** | $0.964 \pm 0.0010$ | $0.909 \pm 0.0037$ | $0.742 \pm 0.0121$ |
| **UniversalTransformer-seq** | $0.969 \pm 0.0004$ | $0.932 \pm 0.0055$ | $0.806 \pm 0.0153$ |
| **LSTM** | $\mathbf{0.977} \pm 0.0001$ | $\mathbf{0.970} \pm 0.0003$ | $\mathbf{0.928} \pm 0.0007$ |

**Task and Models.** We study the performance of LSTMs and variants of Transformers on the task of predicting number-agreement between subjects and verbs in English sentences. We investigate the quality of the solutions they converge to when they are trained independently and when they are trained through distillation. We use the subject-verb agreement dataset of Linzen et al. (2016), for which the size of the training set is ∼121k examples and the size of the test set is ∼1m. Succeeding at this task is a strong indicator that a model can learn syntactic structures and is therefore proposed by Linzen et al. (2016) as a proxy for assessing the ability of models to capture hierarchical structure in natural language. It is shown that RNNs have better inductive biases to learn this compared to standard Transformers (Tran et al., 2018; Dehghani et al., 2019). In this task, examples are grouped into different levels of difficulty based on the number of "agreement attractors"[3], and distance between the verb and its subject. Hence, we report both micro accuracy ($\mu-$Accuracy) and macro accuracy over different groups in terms of distance ($\mathcal{D}-$Accuracy) and numbers of attractors ($\mathcal{A}-$Accuracy).

Similar to Tran et al. (2018), we follow two setups: 1) when the learning objective is next word prediction, i.e., language modeling (LM); 2) when we directly optimize for predicting the verb number, singular or plural, i.e., classification. In the LM setup, we look at the probabilities predicted when the target of the prediction is the verb of interest, and see whether the probability of the correct form of the verb is higher than the other form (singular vs plural). In the classification setup, the input to the model is a sentence up to the position of the verb of interest and the model predicts whether the verb at that position is singular or plural.

In the LM setup, we employ two unidirectional LSTMs with different sizes, *LSTM* and *Small LSTM*, and two Transformers, *Transformer* and *Small Transformer*. In this setup, corresponding LSTMs and Transformers have roughly the same number of parameters. In the classification setup we compare the following models: (1) a standard unidirectional LSTM (*sequentiality + memory bottleneck + recursion*) (2) Transformer: Transformer encoder with a class token (CLS) for classification, BERT (Devlin et al., 2019) style, (3) Transformer-seq: Transformer encoder with future masking where the classification is done using the representation of the last token[4] (*sequentiality*), (4) UniversalTransformer-seq: Universal Transformer (Dehghani et al., 2019) encoder, in which the parameters are shared in depth, with future masking (*sequentiality + recursion*). Appendix H provides more details on the architectures.

## 2.1 THE IMPORTANCE OF RECURRENT INDUCTIVE BIAS

In this section, we report results without distillation that illustrate the merits of the recurrent inductive bias. Table 1 shows the performance of the models when trained with the LM objective. A first important observation, in line with the results of Tran et al. (2018), is that LSTMs achieve better accuracy on the subject-verb agreement task compared to Transformers. Even for instances of Transformer language models that achieve better (lower) perplexity, the accuracy on this task is worse compared to LSTM instances.

---

[3] Agreement attractors are intervening nouns with a different number than the number of the subject. E.g., given the input "The **keys** to the cabinet (is?/are?).", the word "cabinet" is an agreement attractor.

[4] Note that future tokens are masked out by default when using a transformer in the decoder mode, e.g., in LM setup.

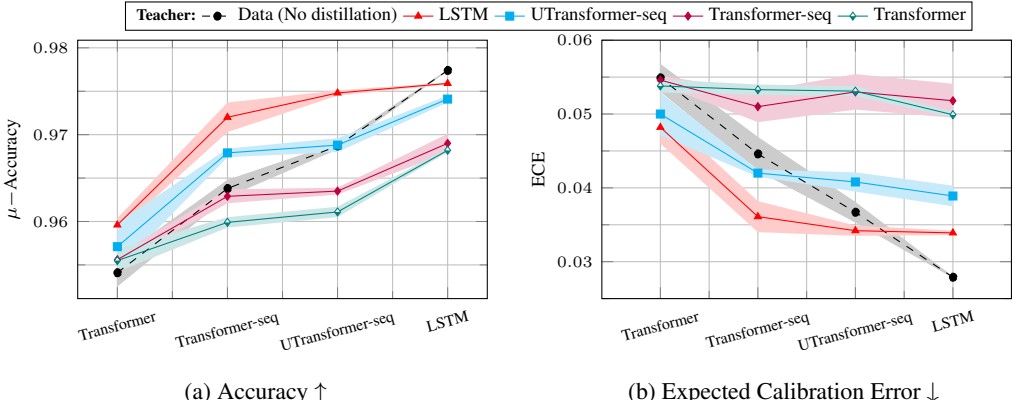

(a) Accuracy ↑            (b) Expected Calibration Error ↓

Figure 3: Performance (mean±std over 4 trials) of models with different inductive biases trained independently or using KD with different teachers.

Since both models achieve good scores on the training set (Appendix D), this suggests that LSTMs better capture relevant patterns, such as the hierarchical structure of the input, which leads to better generalization on this task.

Figure 2 illustrates the accuracy versus perplexity of several instances of each model, in the LM setup. Note that although perplexity is an indicator of how well the model is optimized given the objective function, the accuracy is the metric that matters and shows models' generalization in the subject-verb agreement task. Later, we will show how using KD the behavior of Transformers changes in terms of accuracy versus perplexity and become more similar to LSTM teachers.

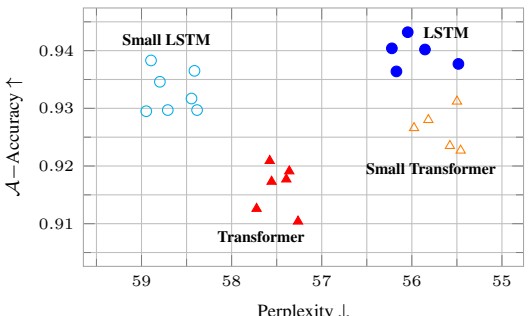

Figure 2: $\mathcal{A}-$Accuracy vs perplexity (high to low from left to right) for language models of different architectures and sizes.

There is another interesting observation in Figure 2. In this plot, for each model, we have two different settings: large and small variants, as measured by the number of trainable parameters. More parameters for a model, given a fixed architecture, means richer hypothesis spaces. We can see that while for the LSTMs, increasing the size of the model results in better performance, for the Transformers increasing the number of parameters results in a worse performance. This aligns with the bias-variance trade-off argument that when using a model with weaker biases for the task at hand, in this case Transformers, if we fix the amount of data, richer hypothesis spaces may hurt the generalization because they increase variance. In contrast, adding more capacity leads to better accuracy in LSTMs as their stronger inductive biases control the generalization error.

In Table 2 we show the results of models trained on the classification objective. We compare LSTM with variants of Transformers with different inductive biases. The table shows that similar to the LM results, LSTM achieves the best performance. Interestingly, comparing all four models, we find that the performance steadily increases as more aspects of the recurrent inductive bias are included. This is illustrated in Figure 3a, with the filled circles on the black, dashed line (no distillation).

As another indicator of the quality of the solutions that different models converged to in the classification setup, we look into their confidence calibration. Confidence calibration captures how well the likelihood (confidence) of the prediction of the model predicts its accuracy (Guo et al., 2017). For a well-calibrated model, if we bin the confidence scores and compute the accuracy for each bin, the accuracies are perfectly correlated with the confidence values. The Expected Calibration Error (ECE) is computed as the distance between the calibration curve of the model and the perfect calibration curve (DeGroot & Fienberg, 1983). In Figure 3b, we plot the ECE (Guo et al., 2017) of the models in the classification setup, with the filled circles on the black dashed line (no distillation). In line with the trends in the performances of these models, the expected calibration error decreases as we move from Transformer toward LSTM.

Table 3: Performance (mean±std over 4 trials) of different LSTM and Transformer models with LM objective when we apply pure distillation with $\tau = 1$.

| Student Model | | Teacher Model | | | |
|---|---|---|---|---|---|
| | | **LSTM** | **Small LSTM** | **Transformer** | **Small Transformer** |
| **LSTM** | $\mathcal{A}-$Accuracy ↑ | $0.9275 \pm 0.0017$ | $0.9310 \pm 0.0013$ | $\mathbf{0.9083} \pm 0.0044$ | $\mathbf{0.9257} \pm 0.0027$ |
| | perplexity ↓ | $59.45 \pm 0.0191$ | $60.92 \pm 0.0185$ | $60.01 \pm 0.0328$ | $58.65 \pm 0.0036$ |
| **Small LSTM** | $\mathcal{A}-$Accuracy ↑ | $0.9224 \pm 0.0024$ | $0.9272 \pm 0.0026$ | $0.8985 \pm 0.0057$ | $0.9157 \pm 0.0020$ |
| | perplexity ↓ | $62.52 \pm 0.1071$ | $63.44 \pm 0.0272$ | $63.45 \pm 0.0644$ | $61.62 \pm 0.0619$ |
| **Transformer** | $\mathcal{A}-$Accuracy ↑ | $\mathbf{0.9296} \pm 0.0029$ | $\mathbf{0.9316} \pm 0.0012$ | $0.8956 \pm 0.0018$ | $0.9195 \pm 0.0015$ |
| | perplexity ↓ | $\mathbf{57.03} \pm 0.0092$ | $59.09 \pm 0.0126$ | $\mathbf{57.67} \pm 0.0091$ | $\mathbf{56.64} \pm 0.0352$ |
| **Small Transformer** | $\mathcal{A}-$Accuracy ↑ | $0.9201 \pm 0.0018$ | $0.9233 \pm 0.0011$ | $0.8827 \pm 0.0027$ | $0.9131 \pm 0.0014$ |
| | perplexity ↓ | $57.84 \pm 0.0269$ | $59.73 \pm 0.0166$ | $58.44 \pm 0.0354$ | $57.16 \pm 0.0087$ |

Table 4: $\mu-$Accuracy ↑ (mean±std over 4 trials) of different LSTM and Transformer models with classification objective when we apply pure distillation with $\tau = 1$.

| Student Model | Teacher Model | | | |
|---|---|---|---|---|
| | **Transformer** | **Transformer-seq** | **UTransformer-seq** | **LSTM** |
| **Transformer** | $0.9555 \pm 0.0013$ | $0.9556 \pm 0.0006$ | $0.9571 \pm 0.0027$ | $0.9596 \pm 0.0008$ |
| **Transformer-seq** | $0.9599 \pm 0.0006$ | $0.9629 \pm 0.0008$ | $0.9679 \pm 0.0005$ | $0.9720 \pm 0.0017$ |
| **UTransformer-seq** | $0.9611 \pm 0.0006$ | $0.9635 \pm 0.0004$ | $0.9688 \pm 0.0008$ | $0.9748 \pm 0.0003$ |
| **LSTM** | $\mathbf{0.9682} \pm 0.0002$ | $\mathbf{0.9690} \pm 0.0011$ | $\mathbf{0.9741} \pm 0.0004$ | $\mathbf{0.9759} \pm 0.0001$ |

In summary, the results from this section support the importance of recurrence for solving this task (Tran et al., 2018; Dehghani et al., 2019). Additionally, as shown in Figures 3a and 3b, we find a decreasing trend in the variance of the models, i.e., adding more inductive biases to the models decreases their variance. This is empirical evidence that supports the relation between variance of the solutions a model converges to and its inductive biases.

## 2.2 TRANSFERRING THE EFFECT OF RECURRENT INDUCTIVE BIAS

In this section, we show distilling knowledge from LSTM to Transformer can close the gap between their performance by pushing the Transformer to converge to solutions more similar to LSTM's.

Table 3 and Table 4 summarize the distillation results, when the training objective is language modeling and classification respectively. A first general observation is that, for these tasks and setups, distilling a model into an identical model could result in a decrease in the performance. Note that whether self-distillation results in improved performance could potentially depend on many different factors such as the architecture of the model, optimization algorithm, and details of the distillation process (Furlanello et al.,

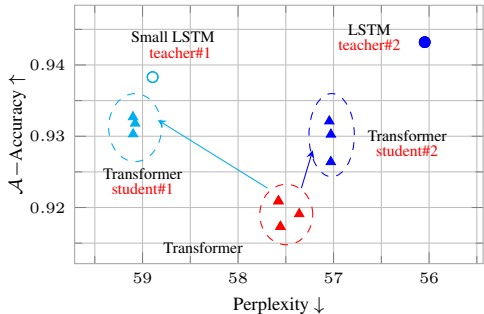

Figure 4: $\mathcal{A}-$Accuracy ↑ vs perplexity ↓ (high to low from left to right) for student Transformer with LM objective. In this figure the triangle marks indicate Transformer models, and circles indicate the LSTM teachers used to train the student Transformers.

2018; Mobahi et al., 2020). Despite no significant changes in the performance with self-distillation, we can improve the performance of the Transformers through distillation from LSTM teachers.

To check whether this improvement is due to the transfer of the effect of inductive biases through distillation and whether distillation helps students to converge to solutions similar to their teachers, we run a series of analyses. In Figure 4 we see how teacher LSTMs pull student Transformers toward solutions with higher accuracy on the subject-verb agreement task in the LM setup. This happens even when the perplexity of the student Transformer is higher (worse) than the independent Transformer.

Figure 3, also shows the effects of distillation on each of the four models we study in the classification setup. In Transformer-based models, we get the most significant improvement both in accuracy and ECE when the teacher is an LSTM. As the recurrent inductive biases of the teacher get weaker, the amount of improvement in the performance of student models decreases. Figure 5 shows the effect of KD on the calibration, given a student Transformer and an LSTM teacher.

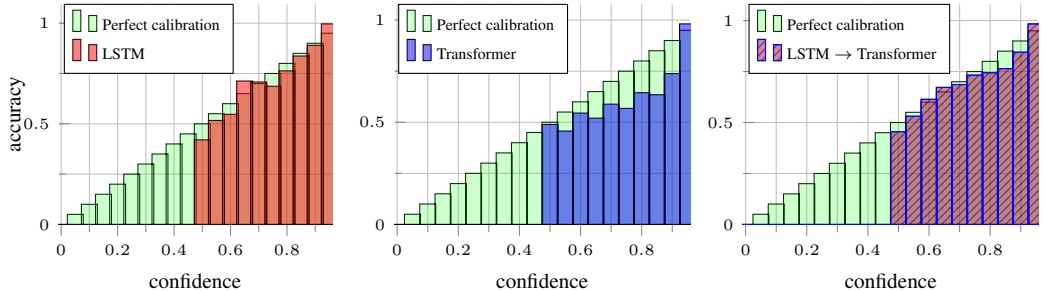

Figure 5: Calibration plots for independent and distilled Transformer for the classification setup. Note that since the task is binary classification, accuracy for confidences lower than 0.5 is not defined.

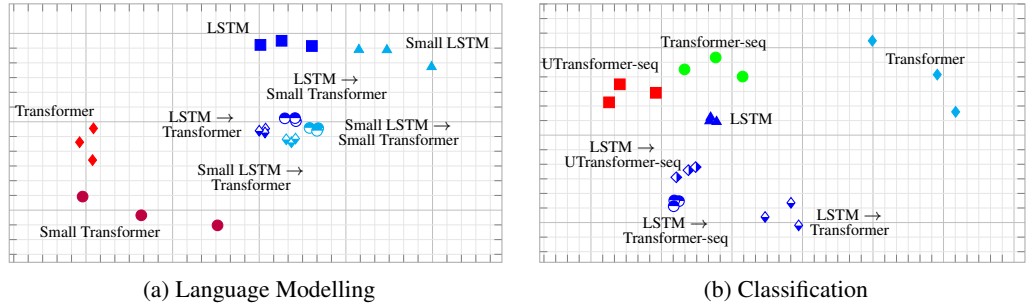

(a) Language Modelling  (b) Classification

Figure 6: 2D projection of representational similarity of the activations from the penultimate layers for 1000 examples from the validation set (check Appendix B for more details). We use the notation of $a \rightarrow b$ to refer to the student model $b$ distilled from teacher model $a$.

**Is the improvement in calibration merely the product of using soft targets?** Mueller et al. (2019) shows training neural networks with soft targets (e.g. through label smoothing) results in models that are better calibrated. On the other hand, KD has a regularization effect similar to label smoothing (Yuan et al., 2019; Tang et al., 2020). Given the lack of significant improvement in ECE in the self-distillation experiments (Figure 3b), it is more likely that the cause of the improvement in ECE when distilling LSTMs into Transformers is beyond the label smoothing effect of KD.

To further explore and better understand the effects of KD, we compare the internal representations of these models besides their final output. Figure 6 shows the 2D projection of the relational similarity of representations[5] (Laakso & Cottrell, 2000) from the penultimate layer of the models (see Appendix B for details). We see that, in the LM setup, the internal representations of student Transformers that are distilled from LSTMs are structured differently compared to independent Transformers and are more similar to the LSTM models. For the classification objective, we also see that the distilled models are further away from their independent versions. This supports the idea that the effect of distillation goes beyond the output of the models and their final performances.

## 3    DISTILLING CNNS INTO MLPS

To evaluate the robustness of our findings on the transfer of inductive biases through KD, we performed a second case study, using different neural architectures and a different task. We use convolutional neural networks (CNN) vs. multilayer perceptrons (MLP) as two families of models with different inductive biases. CNNs are the de facto choice for processing data with grid-like topology. Sparse connectivity and parameter sharing in CNNs make them an effective and statistically efficient architecture. The particular form of parameter sharing in the convolution operation makes CNNs equivariant to translation (Goodfellow et al., 2016). Note that, we can view CNNs as MLPs with an infinitely strong prior over their weights, which says that first of all the weights for each hidden unit are identical to the weights of its neighbor with a shift in space, second, the weights out of the spatially continues receptive field assigned to each hidden unit are zero.

**Task and Models.** We study CNNs and MLPs in the context of the Corrupted-MNIST dataset (MNIST-C) (Mu & Gilmer, 2019), which aims at benchmarking out-of-distribution robustness. We

---

[5]Note that the relational similarity captures the similarity of the structures, not the absolute values.

Table 5: Accuracy and Expected Calibration Error (mean±std over 4 trials) of CNN and MLP trained independently on MNIST and evaluated on MNIST, MNIST-Scaled and MNIST-Translated.

| (a) Accuracy | | | | (b) Expected Calibration Error | | | |
| --- | --- | --- | --- | --- | --- | --- | --- |
| Model | MNIST | Scaled | Translated | Model | MNIST | Scaled | Translated |
| **CNN** | **0.992** ± 0.0009 | **0.962** ± 0.0021 | **0.981** ± 0.0003 | **CNN** | **0.011** ± 0.0006 | **0.060** ± 0.0044 | **0.028** ± 0.0016 |
| **MLP** | 0.985 ± 0.0011 | 0.794 ± 0.0154 | 0.373 ± 0.0151 | **MLP** | 0.015 ± 0.0006 | 0.175 ± 0.0081 | 0.564 ± 0.0091 |

Table 6: Accuracy and Expected Calibration Error (mean±std over 4 trials) of CNN and MLP trained with pure distillation with $\tau = 5$, on MNIST and evaluated on MNIST, MNIST-Scaled and MNIST-Translated.

(a) Accuracy

| Student Model | MNIST | | Scaled | | Translated | |
| --- | --- | --- | --- | --- | --- | --- |
| | **CNN** | **MLP** | **CNN** | **MLP** | **CNN** | **MLP** |
| **CNN** | **0.991** ± 0.0004 | **0.990** ± 0.0007 | **0.951** ± 0.0046 | **0.955** ± 0.0065 | **0.978** ± 0.0003 | **0.976** ± 0.0012 |
| **MLP** | 0.988 ± 0.0005 | 0.985 ± 0.0015 | 0.904 ± 0.0073 | 0.839 ± 0.0096 | 0.510 ± 0.0148 | 0.395 ± 0.0069 |

(b) Expected Calibration Error

| Student Model | MNIST | | Scaled | | Translated | |
| --- | --- | --- | --- | --- | --- | --- |
| | **CNN** | **MLP** | **CNN** | **MLP** | **CNN** | **MLP** |
| **CNN** | 0.014 ± 0.0004 | **0.013** ± 0.0005 | **0.068** ± 0.0043 | **0.054** ± 0.0063 | **0.033** ± 0.0006 | **0.030** ± 0.0016 |
| **MLP** | **0.013** ± 0.0004 | 0.015 ± 0.0012 | 0.109 ± 0.0053 | 0.155 ± 0.0079 | 0.432 ± 0.0136 | 0.555 ± 0.0038 |

train the models on the original MNIST training set and evaluate them on the Translated and Scaled MNIST test sets from MNIST-C. In this scenario, the inductive biases of CNNs help them generalize better than MLPs. Our CNN architecture is a stack of convolutions and pooling layers. Combining convolution and pooling over spatial regions results in invariance to translation. To have CNNs that can learn to be invariant to other transformations like changes in the scale, we can use cross-channel pooling (Goodfellow et al., 2013), where we pool over separately parametrized convolutions that have learned to detect different transformed versions of the same underlying features. Our MLP is simply a stack of fully-connected layers. More details on the architectures are in Appendix H.

## 3.1 THE IMPORTANCE OF TRANSLATION EQUIVARIANCE.

Table 5 presents the accuracy and ECE of CNNs and MLPs when trained independently. All models are trained on the original MNIST training set and tested on the *Scaled* and *Translated* sets from MNIST-C. Even though CNNs' accuracy and ECE on the original MNIST test set are only slightly better than MLPs (.992 vs .985), there is a rather large gap between their performances on the Scaled (.962 vs. .794) and Translated (.981 vs. .373) test sets. This is as expected since the inductive biases of CNNs make them suitable for these types of generalizations. Moreover, the variance of the results from the CNNs is much less compared to MLPs. This is due to the fact that different instances of a model with stronger inductive biases are more likely to converge to solutions that belong to the same basin in the loss landscape (Neyshabur et al., 2020) (See Appendix C for more analysis).

## 3.2 BETTER OUT OF DISTRIBUTION GENERALIZATION WITH KD.

Table 6 shows that distillation from a CNN into an MLP improves both accuracy and ECE for all three test sets, decreasing the gap for the Scaled test set (.904 vs. .794 without KD), and much more improvement on the performance on the Translated test set (.510 vs. .373 without KD). We also see a lower variance in the performance of MLP models that are trained through KD with CNN teachers.

We further compare the results of all possible pairs of models as teachers and students, to take into account different effects of KD that can potentially improve the performance of the student model. Although self-distillation results in a slightly better performance in MLPs, perhaps due to the regularization effect of distillation (Mobahi et al., 2020; Tang et al., 2020), the improvement in the performance of MLPs with an MLP teacher is much less compared to when the teacher is a CNN. Regardless of the teacher (MLP or CNN), KD results in slightly lower performances in student CNNs compared to CNNs trained independently (similar to results of an LSTM student in test case 1).

Furthermore, in Figure 7, we compare the relational similarity of the representations from penultimate layers of independently trained CNNs and MLPs as well as their distilled ones. First of all, as expected based on our assumptions about the inductive biases of these models, MLPs have more variance than CNNs. Second, distilling from a CNN to an MLP results in representations that are more similar to the representations learned by CNNs, while this is not the case with MLPs as teachers

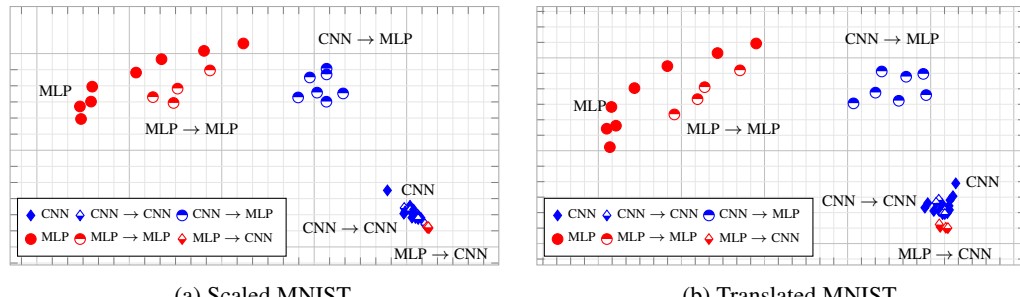

(a) Scaled MNIST            (b) Translated MNIST

Figure 7: 2D projection of representational similarity of the activations from the penultimate layers for all examples from the test set (check Appendix B for more details). We use the notation of $a \to b$ to refer to the student model $b$ distilled from teacher model $a$.

and CNNs as students. Moreover, for both CNNs and MLPs, self-distillation does not significantly change the representations they learn.

Finally, we compare the paths the models follow during training until they converge to a solution. To plot the training path of a model, we compute the pairwise representational similarity between different stages of training of the model. Figure 1, illustrates the training path for an independent MLP, an independent CNN, and an MLP that is distilled from a CNN. While MLP and CNN seem to have very different behavior during training, the student MLP with a CNN as its teacher behaves differently than an independent MLP and more similar to its teacher CNN. This is interesting, in particular, since the student model is only exposed to the final solution the teacher has converged to and no information about the intermediate stages of training is provided in the offline KD.

## 4 CONCLUSIONS

The *no free lunch theorem* states: for any learning algorithm, any improvement on performance over one class of problems is balanced out by a decrease in the performance over another class (Wolpert & Macready, 1997). Neural networks with different architectures have different inductive biases and this is reflected in their performance across different tasks. In this paper, we investigate the power of KD to enable benefiting from the advantages of different models at the same time. First, we demonstrate how inductive biases arising from different architectural choices affect the generalization behavior of the models we study. We further show that when a model has the right inductive bias, we can transfer its knowledge to a model that lacks the needed inductive bias and indicate that solutions that the student model learns are not only quantitatively but also qualitatively reflecting the effects of the inductive biases of the teacher model.

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

## A    KNOWLEDGE DISTILLATION IN NEURAL NETWORKS

Knowledge Distillation is a technique that transfers knowledge from one model to another (Hinton et al., 2015). Hinton et al. (2015) suggest that the power of KD is mostly in being able to transfer the useful information that is embedded in the soft targets of the teacher model, e.g., the relation between the output classes as captured by the teacher model. This is often referred to as *dark knowledge*. Phuong & Lampert (2019) studies KD from a theoretical point of view in a simplified setting where the task is a binary classification, and teacher and student are linear models. They attribute the success of distillation to three main factors: (1) data geometry, (2) optimization bias, and (3) strong monotonicity. And more recently Tang et al. (2020), conduct extensive analysis and identify three sources for why KD helps: (1) label smoothing, (2) example re-weighting based on teacher's confidence, and (3) prior knowledge of optimal output layer geometry.

The most well-known use of KD is to compress a large, unwieldy model or an ensemble model into a smaller model. Empirically, many people have found that bigger models are easier to train (often explained with the 'lottery ticket hypothesis' (Frankle & Carbin, 2019)); KD makes it possible to distill the knowledge in the large model into a much smaller model, and thus in some sense offer the best of both worlds (Buciluǎ et al., 2006; Hinton et al., 2015; Srinivas & Babu, 2015). Distilling knowledge from a very big model or an ensemble of models with similar or heterogeneous architectures that are trained on the same or different tasks into a single model with much fewer parameters can lead to similar or sometimes even better performance compared to the teachers (Luo et al., 2019; Liu et al., 2019; Hinton et al., 2015; Tan et al., 2019; Kim & Rush, 2016).

Previous work has examined the effectiveness of KD in different settings: where the teacher is bigger than the student, but both have similar building blocks (Hinton et al., 2015; Kim & Rush, 2016; Sanh et al., 2019); where teacher and student are of similar size and architecture (Furlanello et al., 2018; Freitag et al., 2017); or where the student and teacher have fundamentally different architectures (Frosst & Hinton, 2017; Tang et al., 2019; Luo et al., 2019; Ahn et al., 2019).

KD has also been proposed as an interpretation technique, where the knowledge of a big complex model is distilled into a more interpretable model (Craven & Shavlik, 1995; Craven, 1996; Frosst & Hinton, 2017); Or as a method to compare the capacity and expressiveness of different models (Maheswaranathan et al., 2019; Saxe et al., 2018).

**Offline Distillation** In most cases, KD is applied in an offline setting, i.e., we first train the teacher network and use the trained teacher to train the student, while the parameters of the teacher are fixed. This is the standard distillation process introduced by Buciluǎ et al. (2006); Hinton et al. (2015). We apply this setup in our experiments since it is the most common approach. There are other possible settings for KD, e.g. online distillation, where teacher and student models are trained simultaneously.

**Distillation Loss** There are several different ways of computing the distillation loss: using only the output of the teacher or taking intermediate layers into account as well (Anil et al., 2018; Ahn et al., 2019; Sun et al., 2019; Park et al., 2019; Tung & Mori, 2019; Buciluǎ et al., 2006; Hinton et al., 2015). Potentially, using these alternative losses could lead to transferring different kinds of knowledge depending on the tasks and the configurations of the models. While it is worth doing a thorough comparison of all these techniques, in this paper we have focused on the most commonly used loss introduced by Hinton et al. (2015), which is based on the Kullback-Leibler divergence between output distributions of the teacher, i.e., soft targets, and the output distributions of the student. The output distributions of the teacher and student model, $P_t$ and $P_s$, are computed similarly, with Equation 1.

$$\frac{\exp(z_i/\tau)}{\sum_j \exp(z_j/\tau)}, \tag{1}$$

where $\tau > 1$ is the softmax temperature and $z$ is the logits from the model.

The distillation loss is: $\mathcal{H}(P_t, P_s)$, where $\mathcal{H}$ is the cross entropy loss and is computed as:

$$\mathcal{H}(P_t, P_s) = -\sum_x P_t(x) \log P_s(x) \tag{2}$$

When KD is applied as a means for model compression, it is common to compute the total loss as a mixture of distillation loss and actual loss. Since, our focus in this paper is on how much the student model can learn from the teacher model, in our experiments we use pure distillation.

## B    VISUALISATION OF REPRESENTATIONAL SIMILARITY OF THE ACTIVATIONS FROM THE PENULTIMATE LAYER

To compare and visualize the state of $m$ different models to each other (at convergence or any stage of training), we propose using representational similarity (Laakso & Cottrell, 2000; Abnar et al., 2019) of the activations from their penultimate layer.

Note that representational similarity measures how similar two models learn to represent the data in terms of the global "relations" between all the data points, not local example-by-example similarity. In fact, the "direct" similarity between the activations of the penultimate layers of two models can be quite low, while having high representational similarity. This is because models can keep the relations between data points similar while embedding data into completely different representational spaces.

This is particularly useful when these models do not have the same architecture and their parameter space is not directly comparable. To do so, given a sample set of size $n$ from the validation/test set (e.g. 1000 examples), we feed them to the forward pass of each model to obtain the representation from the penultimate layer of the models. Then, for each model, we calculate the similarity of the representations of all pairs from the sample set using dot product which leads to a matrix of size $n \times n$. We use the samples similarity matrix associated with each model to compute the similarity between all pairs of models. To do this, we compute the dot product of the corresponding rows of these two matrices after normalization and average all the similarities of all rows, which leads to a single scalar. Given all possible pairs of models, we then have a model similarity matrix of size $m \times m$. We then apply a multidimensional scaling algorithm[6] to embed all the models in a 2D space based on their similarities.

The code for projecting the representational similarity of the activations from the penultimate layer to a 2D space can be found in `https://ANONYMIZED`.

## C    DO THE DISTILLED MODELS CONVERGE TO THE SAME BASIN IN THE LOSS LANDSCAPE?

To gain a better understanding of the effect of KD and inductive biases of the models from an optimization point of view, we looked into how different models relate in terms of the solutions they converged to in the loss landscape.

To do so, inspired by the discussion in (Neyshabur et al., 2020), we look into different pairs of models and check if their final solution belong to the same flat basin[7] of the loss landscape or they converged to completely different optima. To do so, given two models, $m_1$ and $m_2$, we take their parameters, $\theta_1$ and $\theta_2$, and evaluate a series of models obtained by linearly interpolating $\theta_1$ and $\theta_2$, with different coefficient, i.e., the parameters of model $m_i$ is computed as $\theta_i = \lambda_i \theta_1 + (1 - \lambda_i)\theta_2$. It has been shown (Neyshabur et al., 2020) that if the converged solutions of $m_1$ and $m_2$ belong to the same flat basin of the loss landscape, the models obtained by linearly interpolating their parameters are well-behaved because they also remain in that basin. However, for two models that converge to different optima and don't share the flat basin of the loss landscape, the liner interpolations do not lead to well behave models.

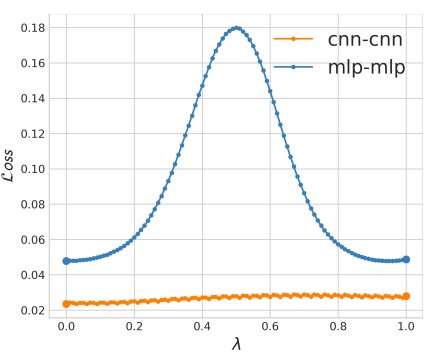

Figure 8: Performance barriers between different instances of MLPs and CNNs (with the same initialization), in terms of loss on the test.

Here, we first, compare different instances of MLPs and CNNs. We train two instances of the same architecture with the same initial state but different random seeds (which would lead to different ordering of training examples, and different dropouts). Figure 8 shows the loss on the test set ($y$ axis) for the two trained instances, as well as models obtained by linear interpolation of the two

---

[6]`https://scikit-learn.org/stable/modules/generated/sklearn.manifold.MDS.html`
[7]Basin refers to areas in the parameter space where the loss function has relatively low values.

models with different $\lambda$s ($x$ axis). In the case of MLPs, there is a large barrier between the two instances, showing that these models, even with the same initialization, will converge to solutions in different basins of the loss landscape. In contrast, for CNNs, their strong inductive biases drive them to converge to the solutions in the same basin, regardless of the stochasticity of the training process. This also supports the higher variance in the results we report for models with weaker inductive biases in §2.2 and §3.2.

Next, we look into the effect of distillation on the diversity of the basins different instances of models converge to. Figure 9 shows the performance barriers of different pairs of MLPs (MLP#1 and MLP#2), when they are trained independently (i.e. when the teacher is data), as well as trained through KD, with an MLP and a CNN model as teachers.

First of all, we observe that two models, initialized similarly but with different random seeds, trained through distillation with the same teacher are likely to converge to the same area in the loss surface (plots (c) and (f)). This happens regardless of the inductive bias of the teacher and student models. Comparing the plots in the diagonal of Figure 9, we can see that for both $CNN \rightarrow MLP$ (plot f) and $MLP \rightarrow MLP$ (plot c) the performance barrier is rather small in contrast to the large barrier between two independently trained MLPs (plot a). This indicates the power of KD to narrow down the search space of the student model and drive it to a particular set of solutions.

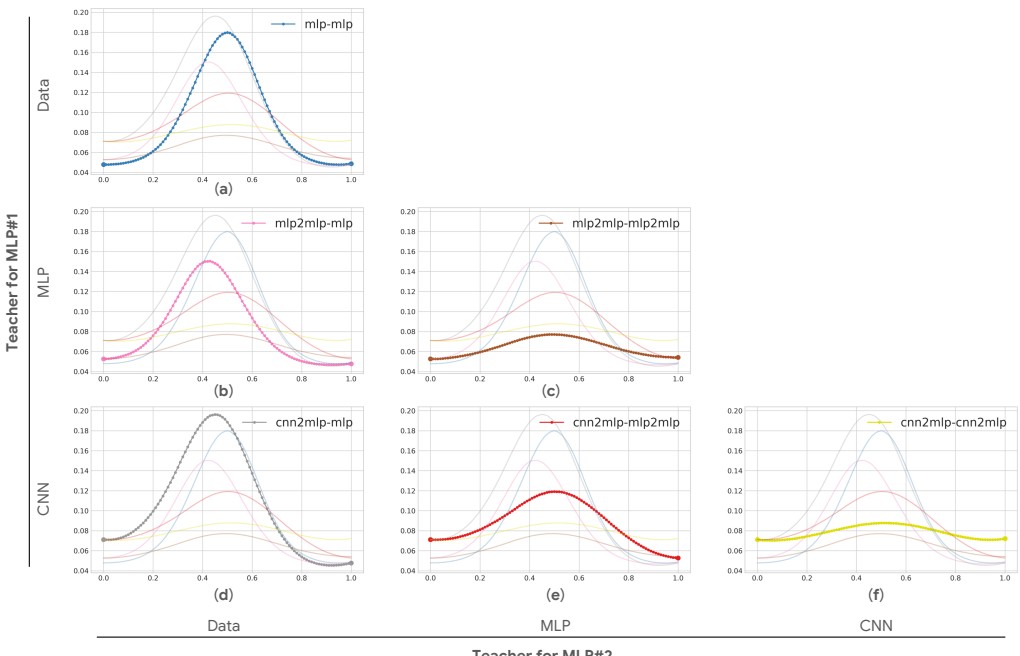

Figure 9: Performance barriers between different instances of MLPs with the same initialization trained independently or through knowledge distillation. Here $y$-axis on each subplot is the value of the loss on the test set and $x$-axis is the value of the interpolation coefficient, $\lambda$. The rows in the figure correspond to the teacher of the instance on the left side (MLP#1) and the columns correspond to the teacher of the instance on the right side of the plots (MLP#2).

Moreover, comparing the distilled instance of a model with an independently trained instance with the same initialization and different random seeds, the first column of Figure 9 (plots (a), (b), and (d)), we see that the distilled instances and independent instances are not in the same basin, regardless of the teacher but the barrier is larger (larger bump in the plots) when the teacher has a stronger inductive bias ($CNN \rightarrow MLP$). Similarly, as depicted in the second and third columns of Figure 9, while models distilled from the same teacher seem to be close in the loss surface (plots (c) and (f)), models distilled from different teachers (plot (e)) seem to be further away (have a larger barrier in between).

## D    PERFORMANCE SCORES ON THE TRAINING DATA

In the paper, for our first test case, we report the performance of LSTM and different Transformer models on the test set, when trained independently and with knowledge distillation. We observe that LSTMs achieve better accuracy on the test set compared to Transformers due to their inductive biases. Here, we also report the performance of all the models, for both classification and LM setup, on the training set, which confirms that Transformer models have enough capacity to achieve good scores on the training data.

This solidifies the narrative that the inductive bias of LSTMs is helping with generalization and rules out, for example, the possibility that LSTMs have a higher capacity or are trained better.

| Model | Perplexity $\downarrow$ | $\mathcal{D}-$**Accuracy** $\uparrow$ | $\mathcal{A}-$**Accuracy** $\uparrow$ |
|---|---|---|---|
| **Transformer** | $29.62 \pm 0.10$ | $0.956 \pm 0.001$ | $0.936 \pm 0.004$ |
| **Small Transformer** | $33.02 \pm 0.05$ | $0.959 \pm 0.001$ | $0.948 \pm 0.005$ |
| **LSTM** | $28.92 \pm 0.08$ | $0.964 \pm 0.003$ | $0.955 \pm 0.003$ |
| **Small LSTM** | $31.03 \pm 0.11$ | $0.964 \pm 0.001$ | $0.952 \pm 0.006$ |

Table 7: Performance (mean±std over 4 trials) of different LSTM and Transformer models trained independently with the LM objective on the training set.

| Model | Train $\mu-$**Accuracy** $\uparrow$ |
|---|---|
| **Transformer** | 99.57 |
| **Transformer-seq** | 99.57 |
| **UniversalTransformer-seq** | 99.66 |
| **LSTM** | 98.62 |

Table 8: Performance (mean±std over 4 trials) of different LSTM and Transformer models trained independently with the classification objective on the training set.

## E    PER-SAMPLE BEHAVIOUR

To compare the models with each other and better understand how distillation affects the student models, we take a closer look at their per sample behavior and investigate if the errors a student model makes are more similar to its teacher's errors. Here, we look into the error overlap of the students and teachers, which reflects their similarity in terms of their behavior per data example. This similarity can be another proxy to measure the similarity of the solutions learned by the models, with and without distillation. Figures 10, 11, and 12 illustrate the error overlap between different models as Venn diagrams when they are trained independently and when we use distillation.

In Figure 10, we observe that when the Transformer and LSTM models are trained independently, two independent LSTMs behave more similarly compared to two Transformers (Figures 10b and 10a). Given a similar number of trainable parameters, i.e., similar capacity for LSTMs and Transformers, this again supports the claim that models with stronger inductive biases converge to more similar solutions (Also shown in Figure 3a).

When we apply KD in a cross-architecture setting, with an LSTM teacher and a student Transformer, Figures 10d and Figure 10c, the student Transformer behaves more similarly to the LSTM teacher and an independent LSTM, compared to the independent version of itself. This confirms that through distillation the way the student model solves the task becomes more similar to the way the teacher model solves the task.

We have similar observations in Figures 11, and 12; where errors of a student MLP are less and more similar to the errors the teacher CNN compared to an independently trained MLP.

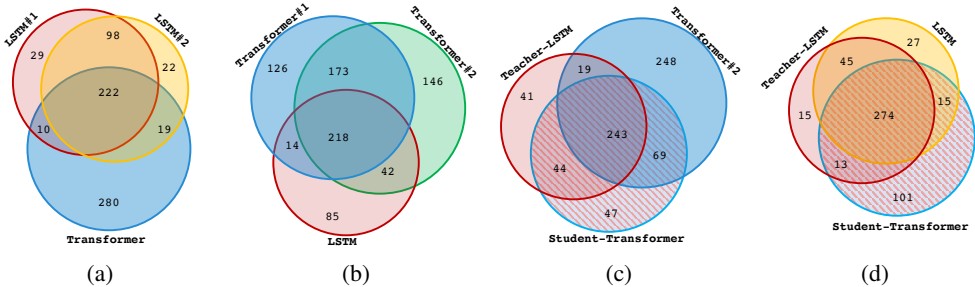

Figure 10: Error overlap for LSTM and Transformer models trained with the classification objective on SVA task. These Venn diagrams show the intersections of the sets of examples miss-classified by the models. In (a) we compare two independent LSTMs (LSTM#1 and LSTM#2) and an independent Transformer; in (b) we compare two independent Transformers (Transformer#1 and Transformer#2) and an independent LSTM; in (c) we compare a student Transformer and a teacher LSTM with an independent Transformer; and in (d) we compare a student Transformer and a teacher LSTM with an independent LSTM.

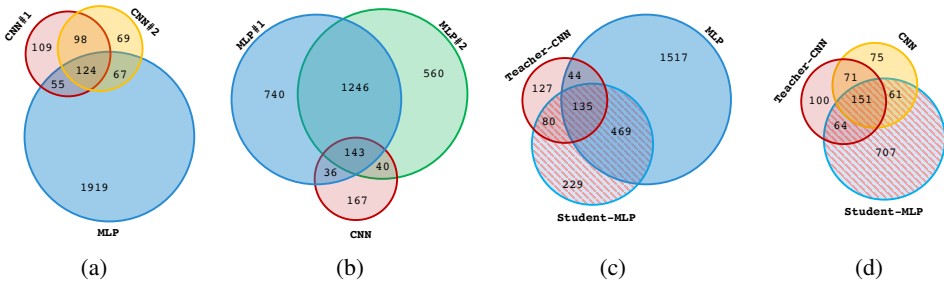

Figure 11: Error overlap for CNN and MLP models trained on MNIST and tested on Scaled-MNIST set from MNIST-C dataset. These Venn diagrams show the intersections of the sets of examples miss-classified by the models. In (a) we compare two independent CNN (CNN#1 and CNN#2) and an independent MLP; in (b) we compare two independent MLP (MLP#1 and MLP#2) and an independent CNN; in (c) we compare a student MLP and a teacher CNN with an independent MLP; and in (d) we compare a student MLP and a teacher CNN with an independent CNN.

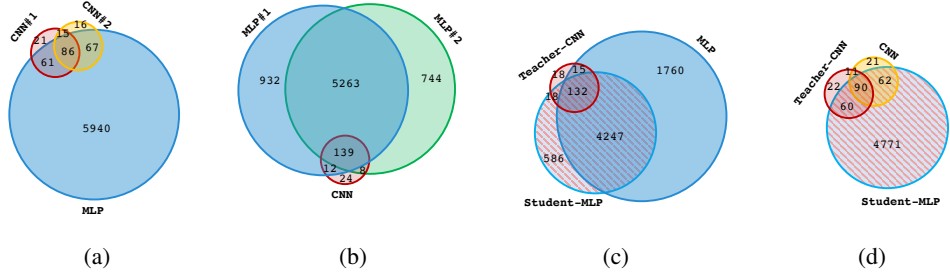

Figure 12: Error overlap for CNN and MLP models trained on MNIST and tested on Translated-MNIST set from MNIST-C dataset. These Venn diagrams show the intersections of the sets of examples miss-classified by the models. In (a) we compare two independent CNN (CNN#1 and CNN#2) and an independent MLP; in (b) we compare two independent MLP (MLP#1 and MLP#2) and an independent CNN; in (c) we compare a student MLP and a teacher CNN with an independent MLP; and in (d) we compare a student MLP and a teacher CNN with an independent CNN.

## F    IMPACT OF THE QUALITY OF THE TEACHER

Here, in an ablation experiment for our second case study, we investigate the impact of the quality of the teacher in the in-distribution set on the generalization of the student in the out-of-distribution set. To do so, given a CNN as the teacher and an MLP as the student, we take snapshots of a CNN model

during different stages of training as teachers with different qualities (we use 9 different teachers). Using each teacher, we train an MLP student.

Figure 13a presents the quality of the different teachers based on different test sets: Vanilla MNIST (in-distribution), Translated MNIST (out-of-distribution), and Scaled MNISt (out-of-distribution). For the CNN models that are trained with ground truth labels on vanilla MNIST, as expected, as the number of training iterations grows, the performance of the model on all the three test sets increases. In Figure 13b, we see that in general, the accuracy of the MLP students follows the same trend, i.e., better CNN teacher results in a better MLP student. Given the results of an independently trained MLP from Table 5a, the benefit of training an MLP via distillation for better generalization on in and out of distribution sets only kicks in when we have a CNN teacher with a quality more than a certain threshold.

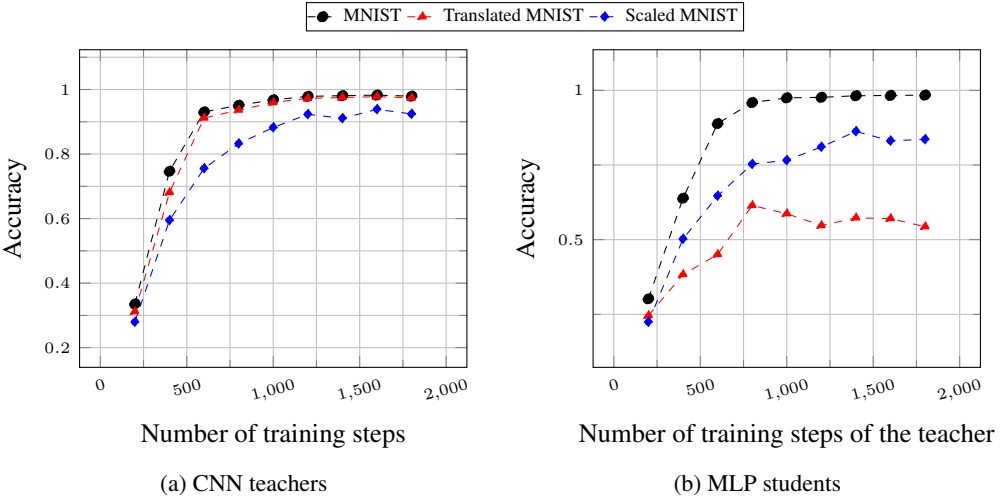

(a) CNN teachers           (b) MLP students

Figure 13: Effect of the quality of the teacher CNNs on the accuracy of the student MLPs. In the left plot, points that share the value on the x-axis represent the quality of a CNN, with respect to different test sets: Vanilla MNIST (in-distribution), Translated MNIST (out-of-distribution), and Scaled MNISt (out-of-distribution). In the right plot, similar to the left plot, points with the same x-value represent the quality of a same MLP model, trained via KD using the teacher on the corresponding place in the left plot, evaluated on the Vanilla, Translated, and Scaled MNIST test sets.

## G   IMPACT OF THE DATASET USED IN THE DISTILLATION STEP

In our experiments in this paper, our focus is on the setups where we use the same dataset that was used to train the teacher model, to transfer its knowledge to the student model.

We use this setup mainly because we want to see how effective is the distillation process to transfer the generalization behavior of the teacher in isolation, as using a different dataset in the distillation step would add another factor. In other words, during the training of the teachers and as well as the students (i.e., distillation step), we only use samples from the in-distribution set to make sure the desired generalization behavior is not apparent from the dataset used for training neither the teacher nor the student models.

In this section, we extend the CNN-MLP experiments on Corrupted-MNIST and look into the performance of the student model, when we use samples from the out-of-distribution set in the distillation step for training the student.

Table 9 presents the result of an MLP student when we use different training sets in the distillation step. We can see that when distilling knowledge from a CNN teacher that is trained on vanilla MNIST, if we use translated or scaled MNIST in the distillation step, the student MLPs achieve relatively high performance on the corresponding test sets, while the performance on the other out-of-distribution set drops compared to when we use vanilla MNIST in the distillation step. To have complementary information for better comparisons, Table 10 shows the accuracies of MLPs when

Table 9: Accuracy of MLPs trained through KD with CNN teachers, where the CNN teachers are trained on in-distribution (vanilla MNIST) training set, while the training set in the distillation step is either in-distortion (first row) or out-of-distribution (second and third rows). Note that during the distillation step, the student do not have access to the ground truth labels from the training set.).

| Distillation Dataset | Test Dataset | | |
| --- | --- | --- | --- |
| | MNIST | Translated MNIST | Scaled MNIST |
| MNIST | $0.99 \pm 0.001$ | $0.51 \pm 0.015$ | $0.90 \pm 0.007$ |
| Translated MNIST | $0.79 \pm 0.015$ | $0.98 \pm 0.001$ | $0.53 \pm 0.015$ |
| Scaled MNIST | $0.79 \pm 0.016$ | $0.30 \pm 0.009$ | $0.98 \pm 0.001$ |

Table 10: Accuracy of MLPs trained with ground truth labels on different splits of the Corrupted-MNIST dataset.

| Training Dataset | Test Dataset | | |
| --- | --- | --- | --- |
| | MNIST | Translated MNIST | Scaled MNIST |
| MNIST | $0.99 \pm 0.001$ | $0.37 \pm 0.015$ | $0.79 \pm 0.015$ |
| Translated MNIST | $0.62 \pm 0.011$ | $0.98 \pm 0.001$ | $0.42 \pm 0.039$ |
| Scaled MNIST | $0.76 \pm 0.014$ | $0.26 \pm 0.006$ | $0.99 \pm 0.001$ |

Table 11: Accuracy of CNNs trained with ground truth labels on different splits of the Corrupted-MNIST dataset.

| Training Dataset | Test Dataset | | |
| --- | --- | --- | --- |
| | MNIST | Translated MNIST | Scaled MNIST |
| MNIST | $0.99 \pm 0.001$ | $0.98 \pm 0.000$ | $0.96 \pm 0.002$ |
| Translated MNIST | $0.99 \pm 0.001$ | $0.99 \pm 0.001$ | $0.97 \pm 0.002$ |
| Scaled MNIST | $0.88 \pm 0.010$ | $0.88 \pm 0.014$ | $0.99 \pm 0.001$ |

they are directly trained on each of these datasets. Interestingly, we observe that when trained through KD the performance of the student MLPs are higher or match the performance of MLPs when trained with ground truth labels, and additionally, they achieve better performance on the other datasets. For example, for the case when we use translated MNIST to train the MLPs, the accuracy of the student MLPs match the accuracy of MLPs trained with the ground labels, while the accuracies on the other two datasets (vanilla and scaled) are higher for the student MLPs trained with CNN teachers.

## H  DETAILED MODELS ARCHITECTURES AND TRAINING SETUP

For the subject-verb agreement task, we study Transformers and LSTMs. In the LM setup, we use two sizes for each architecture: LSTM: two-layer uni-direction LSTM, with a hidden size of 1024. Small LSTM: two-layer uni-direction LSTM, with a hidden size of 512. Transformer: six-layer Transformer decoder with a hidden size of 512 and 8 heads. Small Transformer: Transformer: six-layer Transformer decoder with a hidden size of 256 and 8 heads.

In the classification setup, we employ an LSTM and three variants of Transformer, where the LSTM has a two-layer with a hidden size of 256, and the Transformers have 6 layers, 8 heads, and a hidden size of 128. We use a hidden size of 256 for the UniversalTransformer-seq since its parameters are shared in depth and with the same hidden size as other Transformers, it will have fewer parameters.

On the MNIST-C dataset, we study CNNs and MLPs. Our CNN has two $3 \times 3$ convolutions, followed by a max-pooling layer over spatial dimensions, followed by another $3 \times 3$ convolution and a maxout (max-pooling over channel dimension) layer (Goodfellow et al., 2013). Finally a global averaging is done over spatial dimensions, before the projection layer. The MLP model simply has three fully connected layers.

For training the independent models we use the Adam optimizer (Kingma & Ba, 2014) with exponential decay learning rate scheduler and for the student models in the distillation process, we use Adam optimizer with cosine decay restart (Loshchilov & Hutter, 2017) learning rate scheduler. The

hyperparameters related to the regularization and learning rate schedulers are tuned separately for each model/experiment. For each model, we report the set of hyper-parameters that gives the best average performance across multiple trials with different random seeds for initialization.

## I  CODE

The code for all the analysis and experiments including the input pipelines, models, the details of the hyper-parameter sets used in our experiments are available at `https://ANONYMIZED`, to facilitate the replication of all the experiments.

For the anonymized submission, we remove the links to the repository of the code from the paper and upload the code as a zip file as part of the supplementary materials for our submission.

