# OpenReview forum: "Transferring Inductive Biases through Knowledge Distillation"
_ICLR.cc/2021/Conference — Reject_

### Official Review · AnonReviewer3 · 2020-10-28
**Review of 'Transferring Inductive Biases through Knowledge Distillation'**

**Rating:** 5
**Confidence:** 4

**Review:**

This paper shows that knowledge distillation from a (teacher) model A with an appropriate inductive bias to a (student) model B lacking it can lead to B generalizing better than if B was trained without knowledge distillation (but not as well as A), including out-of-distribution. The authors also show that the resulting learned representations inside B, as well as the shape of the training trajectories, are more like those of A (than those of B without knowledge distillation).

This is not very surprising but is still interesting from the point of view of the understanding of the nature of inductive biases. We already knew that inductive biases (like translation invariance) can be transferred through examples (e.g. by generating data transformations such as translated images), so this paper extends that kind of idea to knowledge distillation to provide the targets for such examples.

Another nice contribution of the paper is the case study of the specific inductive biases of RNNs which transformers lack, decomposed into sequentiality, memory bottleneck and recursion. Not very surprising but the experiments confirm intuitions and expectations which is always useful.

One concern I have is 'so what?' and 'then what?'. Have the authors thought of possible way (in say, future work) to take advantage of that observation? It is not obvious, because if you already have a teacher model A with the right inductive biases for the given task, why would you care about training a student B which is going to be worse than A anyways? Just use A. In addition, unlike for the original motivation of knowledge distillation, we normally expect that B would have MORE capacity than A (because it needs to 'learn' the inductive biases, so one would expect it would not work to choose B much smaller than A, in the sense that the gain would be much smaller, and certainly not as good a model as using A).  We already knew that examples could transfer inductive biases, now we know that knowledge distillation can do it, but why would that be useful?

Experiments where B is much smaller than A would be interesting, because in that case, it might be worthwhile to do the knowledge distillation from a larger but better biased A. Also, the outcome of such experiments would not be apriori obvious (we would expect a gain vs the regular B, but would it be sufficiently interesting to be worth it?).

Another question I would have liked to be studied is about what happens out-of-distribution (OOD). The paper already shows the unsurprising  result that distilling into B from A helps somewhat OOD. It would also be interesting to explore whether taking inputs outside of the training distribution of A as distillation examples when training B would increase the robustness of B OOD.

Minor comments:

- fig 4: caption is insufficient to understand the figure
- the sec 3.2 sentence with 'almost closing the gap' is too strong and needs to be weakened (there is still a significant gap, with almost twice the error with B compared with A)
- the conclusion sentence with 'demonstrate having the right inductive bias can be crucial' should be reformulated, since this is not a new demonstration (and reading it without reading the rest of the paper may give that false impression)

---

> ### Author Response · Authors · 2020-11-18
> **Responses to comments #1, #2 (from reviewer #3)**
>
> We would like to thank reviewer#3 for their valuable feedback. We have updated the paper and address the minor comments mentioned at the end of the review and incorporated suggestions from the reviewer. In the following, we respond to their comments.
>
> ## Response to comment #1
> ```>> One concern I have is 'so what?' and 'then what?'. Have the authors thought of possible way (in say, future work) to take advantage of that observation? It is not obvious, because if you already have a teacher model A with the right inductive biases for the given task, why would you care about training a student B which is going to be worse than A anyways? Just use A. In addition, unlike for the original motivation of knowledge distillation, we normally expect that B would have MORE capacity than A (because it needs to 'learn' the inductive biases, so one would expect it would not work to choose B much smaller than A, in the sense that the gain would be much smaller, and certainly not as good a model as using A).   Experiments where B is much smaller than A would be interesting, because in that case, it might be worthwhile to do the knowledge distillation from a larger but better biased A. Also, the outcome of such experiments would not be apriori obvious (we would expect a gain vs the regular B, but would it be sufficiently interesting to be worth it?).```
>
> Thanks for raising this question. This is an important question and we have tried to make this clear at the beginning of our introduction where we say [“The advantage of KD goes beyond model compression and it can be used to combine strengths of different learning algorithms [...]” and in our conclusion where we say: “The no free lunch theorem states: for any learning algorithm, any improvement on performance over one class of problems is balanced out by a decrease in the performance over another class. [....] In this paper, we investigate the power of KD to enable benefiting from the advantages of different models at the same time.”
> As an instance, the scenarios of your example that we have models A and B, and A has the right inductive bias and thus has better performance, but B is much faster at inference. There is then a clear advantage to use KD to hopefully enable B to learn the solution that A can converge to.  As an example of this, in the classification setup in our first case study, where we use models in the encoder mode, LSTM has the right inductive bias, but due to the sequential nature of it, it can be extremely slow for longer sequences, while the transformer model can be much faster due to the parallelization in processing input tokens, while it lacks the right inductive bias for learning the task in that case study. This is only one example and there are many practical situations where the model B has a benefit over A, which can be orthogonal to the performance (e.g., size of the model as it’s the most common motivation for KD).
> Now that ML researchers work on different algorithms with different properties, research works in the direction of finding ways to have the benefits of multiple learning algorithms in one place can be really beneficial and we believe, our work takes a small step in this direction.
>
> ***
> ## Response to comment #2
> ```>> We already knew that examples could transfer inductive biases, now we know that knowledge distillation can do it, but why would that be useful?```
>
> We would like to emphasize that the inductive bias of a model is defined as preferences of that model independent of the data it observes. We would like to point out that it’s not always possible to learn about a specific aspect (like modeling the hierarchical structure of the input in our first case study) from the data, when the data is limited. Note that no data augmentation can be used in this case to compensate for the lack of enough data. So it’s valuable to have models that have the right inductive bias to learn such a property. And showing that property can be transferred from model A to model B via KD is in particular important in these cases as there is no option for model B to learn that from the data.

---

> ### Author Response · Authors · 2020-11-18
> **Response to comments #3 (from reviewer #3)**
>
> ```>> Another question I would have liked to be studied is about what happens out-of-distribution (OOD). The paper already shows the unsurprising result that distilling into B from A helps somewhat OOD. It would also be interesting to explore whether taking inputs outside of the training distribution of A as distillation examples when training B would increase the robustness of B OOD.```
>
> We have extended this suggestion and added a new set of experiments to the updated version of the paper (please check the Appendix G). We also discuss our findings here.
> We can assume the following cases related to our second case study:
> 1. We know that when training MLP on the translated MNIST (no KD), the MLP can pick up ‘the translation invariance’ property from the observed data. It is to some extent  similar to having data augmentation and if we test on the translated MNIST, we no longer test generalization on a OOD set . [Results are shown in the updated version of the paper, in Appendix G ]
> 2. We also know that training a CNN only on vanilla MNIST, due to its inductive bias, can generalize to the translated test set (as the out of distribution set). [Results already shown in the paper,  Table 5.a]
> 3. Now, per your suggestion, for training an MLP with KD,  if we use translated samples from the OOD as distillation examples and use the CNN as teacher, given the quality of the teacher is really good and the predictions are really close to ground truth in the translation set, this setup will become similar to the setup (1), which is basically augmenting the data and we believe it’s not surprising if MLP learns to generalize to the translation set. In other words, the signal for learning that property comes both from the teacher and the data and it’s easier to learn it. [Results shown in the updated version of the paper, Appendix G]
> 4. However, in our setup, MLP is not exposed to any data (even as distillation examples) with properties of the translated set and it is supposed to learn it merely based on the output of the teacher (which also didn’t observe any sample with properties of the translated set) on in-distribution data. And the fact that the MLP learns to generalize to OOD in this setup sounds much more surprising than (3) and has never been shown before.  [Results already shown in the paper,  Table 6.a]
>
> Appendix G presents a more complete version the experiments above with some additional balations. From our experiment in Appendix G, we can see that when distilling knowledge from a CNN teacher that is trained on vanilla MNIST, if we use translated or scaled MNIST in the distillation step, the student MLPs achieve relatively high performance on the corresponding test sets, while the performance on the other out-of-distribution set drops compared to when we use vanilla MNIST in the distillation step.

---

### Official Review · AnonReviewer2 · 2020-10-29
**Great paper with jarring flaws**

**Rating:** 7
**Confidence:** 4

**Review:**

The paper investigates the oft-overlooked aspect of knowledge distillation (KD) -- why it works. The paper highlights the ability of KD for transferring not just the soft labels, but the inductive bias (assumptions inherent in the method, e.g. LSTM's notion of sequentiality, and CNN's translational invariance/equivariance) from the student so that the student exhibits, to an extent, the teacher's generalization properties as well. The paper explores doing KD between LSTMs and several versions of Transformers (with varying structural constraints) on a subject-verb-agreement dataset, and between CNNs and MLPs on MNIST and corrupted MNIST. Compared to prior work showing that better teacher performance lead to better student performance, this paper also shows that the student's performance on different aspects becomes more similar to the teacher's -- (1) if the teacher is strong on metric A and weak on metric B compared to a student on its own, the student can become stronger on A and weaker on B when distilled using the teacher; (2) if the teacher can generalize well to a separate, previously unseen dataset but the student generalizes poorly on its own, after distillation the student can generalize much better than it can possibly learn to on its own.

Pros:
- Very interesting hypothesis and sheds light on the inner working of KD. (see above)
- Interesting and novel set of experiments. Some (not all) experiments shed light on how the hypothesis seems to be true. (see above)
- Comes up with ways to measure transferred inductive bias, by highlighting different aspects of generalization for a student and comparing with and without distillation.

Cons:
- The writing is very confusing and cryptic, especially the first page until its last paragraph.
    1. The abstract is especially not telling the readers much about what is in the paper. I personally would be confused and skip reading this paper because I thought the paper discusses "can we distill knowledge using knowledge distillation". Inductive bias come in many forms and is not often discussed, and it helps to use examples to tell the story directly, e.g. by mentioning the specific differences between inherent priors in CNNs/MLPs or LSTMs/Transformers in the abstract *and* the first paragraphs of the introduction.
    1. Second page, bold "Second" and "Third" are the same thing.
- Although after extensive thinking I believe the paper is distinct from previous KD analyses, the paper does not itself distinguish its findings enough from what is known in the literature.
    1. Granted it is hard to distinguish the inductive bias transfer aspect of KD versus other aspects of KD, it is hard to experimentally prove it because the field does not quite know what are the aspects of KD that makes it work. But the paper does not do a good job explaining which behavior is certainly due to inductive bias transfer, rather than the behavior can possibly be caused by other hypothesis in the field, such as KD transfer "knowledge" of inter-class relationship, or the effect from soft labels.
    1. Note that the ECE results don't tell readers much. People expect soft labels to help not because they make models better calibrated, but because they boost performance, and it's not clear if people think better calibration leads to better main performance. Even if people do, in Fig 3(b) the ECE improves quite a lot for Transformer student with better teachers, so it is wrong to claim "Given the lack of significant improvement in ECE...".
    1. The CNN/MLP experiment only has tasks that CNNs outperform MLPs. It would make it more interesting to see a task where the MLP outperforms CNN, e.g. a made up task whose ground truth is the xor of a few pixel positions, which could be hard for CNNs while easy for MLPs.
- Relatively small number of datasets. Just two datasets and two sets of networks is not very convincing to claim these findings generalize to other architectural changes.
- Experiments are sometimes not apple-to-apple comparisons. And some experiments are not convincing or irrelevant.
    1. The MNIST experiments only have two networks, and arguably CNN is absolutely better than MLP. It would make the point clearer if a worse CNN is used such that the MNIST-vanilla performance is the same as the MLP, and show improved generalization results on MNIST-C.
    1. Figure 1 does not tell readers much, because latent representation can be both inductive bias and regular representation power, and we already know that KD can improve the student's representation power. Same for the third bullet point in page 2.
    1. Suspect of cherry-picking results from which loss (LM or classification) to show. Figures 2,4 are experiments using the LM loss, and Figures 3,5 are using the classification loss, without giving a clear explanation why.

Summary:
Given the interesting hypothesis and set of experiments, I think the community can benefit from this paper's findings in understanding KD so we use it more wisely, or at least generate more discussions of why KD is working. Despite the relatively unclear writing of the introduction and some experiments being unconvincing, the impact of the paper still outweighs these flaws.

=============================
Update
While I agree in principle with Reviewer 1 that this paper has jarring flaws in writing and the rebuttal version does not adequately address it, I disagree that the writing warrants such a low score. I have seen worse papers with outrageous claims (e.g. try to claim significance with p=0.1) and I would not give those a 2. I would also disagree with R1 that there is no interesting result in this paper, because there is no prior work I know that even considers how distilled models generalize like their teacher.
If I were to grade this paper based on different aspects, the originality and significance would be both 9's, quality a 6 due to experiment issues and careless generalization, and clarity a 3-4 due to unclear motivation in the abstract/early intro and poor differentiation from prior work in terms of experiment design and analysis.
That said, the rebuttal did not change my mind that the writing probably will not be improved enough post-rebuttal, I would thus not be able to consider this a top paper despite the interesting observations.

---

> ### Author Response · Authors · 2020-11-18
> **Responses to comments #1, #2, #3 (from reviewer #2)**
>
> We’d like to thank the reviewer for the kind words, and thoughtful comments about the paper. We reply to the comments from the reviewer. We’d also like to appreciate that while the reviewer raised their concerns about the paper and provided suggestions, they recognized the contributions of the paper and potential impact of the findings in this paper.
> ***
> ## Response to comment #1
> ```>> 1. The abstract is especially not telling the readers much about what is in the paper. I personally would be confused and skip reading this paper because I thought the paper discusses "can we distill knowledge using knowledge distillation". Inductive bias come in many forms and is not often discussed, and it helps to use examples to tell the story directly, e.g. by mentioning the specific differences between inherent priors in CNNs/MLPs or LSTMs/Transformers in the abstract and the first paragraphs of the introduction.```
>
> Thanks for your feedback and for the great suggestions. We modified the abstract and the introduction to better reflect what is done in the paper.
> ***
> ## Response to comment #2
> ```>> 2. Second page, bold "Second" and "Third" are the same thing.```
>
> We agree that the second and third items are to some extent hard to distinguish, however they meant to state different things.  We updated this part of the introduction and presented a new structure to spell out the findings of the paper  in the updated version to make this more clear.
> Here  we also bring a short summary. When using KD,  we compare different models in different setups: (1) when trained without KD, but directly from the data, (2) when trained with KD using a teacher with a similar architecture to the student, i.e. self-distillation, and (3) when trained with KD using a teacher with a different architecture that has better inductive biases that the student.
> In the second point:
> “Second, we show that KD is a powerful technique [...]”.
> We contrast setups (2) and (3) with setup (1) and present how KD affects the student, regardless of the properties of the teacher.
> In the third point  (and it’s sub-items):
> “Third, we demonstrate that, when distilling the knowledge from a model with stronger inductive bias [...]”.
> We focus on contrasting setup (3) with setups (2) and (1),  and show the effect of KD, when we have a teacher with better inductive bias.
> Thanks again for your feedback on this, which led to a much better structure for this part of the  introduction.
> ***
> ## Response to comment #3
> ```>> Although after extensive thinking I believe the paper is distinct from previous KD analyses, the paper does not itself distinguish its findings enough from what is known in the literature.```
>
> Thanks for the comment. In the updated version of the paper, we made this more clear that how the experiments in the paper are designed to study a different aspect from what is already investigated in the literature.  However, for the sake of completeness, we also reply to each point here (in the responses to comment #4 and comment #5).

---

> > ### Comment · AnonReviewer2 · 2020-11-24
> > **Additional feedback on #1,2**
> >
> > # 1
> > The new version reads better, but it is still obscured. The difference of MLP and CNN, transformer and LSTM are not stated in page 1. I'm more suggesting that you explain "why you do this" not "how you do this" (as with the added abstract content). I'll just say page 2 does a better job of explaining the paper than page 1.
> >
> > # 2
> > The difference is subtle and the new version contrasts them better. Not a big issue, but I still think they can be merged with point 2 becoming the first (extra) bullet point of point 3.

---

> ### Author Response · Authors · 2020-11-18
> **Responses to comments #4, #5 (from reviewer #2)**
>
> ## Response to comment #4
> ```>> 1. Granted it is hard to distinguish the inductive bias transfer aspect of KD versus other aspects of KD, it is hard to experimentally prove it because the field does not quite know what are the aspects of KD that makes it work. But the paper does not do a good job explaining which behavior is certainly due to inductive bias transfer, rather than the behavior can possibly be caused by other hypothesis in the field, such as KD transfer "knowledge" of inter-class relationship, or the effect from soft labels.```
>
> One of the key differences between our paper and previous works that investigated KD and how it helps is that we target the setup in which the teacher has a different architecture (which has the right inductive bias for the task at hand), while in most of the previous works are focused on KD in a setup where teacher and student are different in terms of model size. Then the important point here becomes, what is the difference between the way KD helps when teacher and student have different inductive bias, compared to when we apply KD on a teacher and student with similar architecture. In all our experiments, we have three cases: (1) when the model is trained independently from the training data (e.g. Data -> MLP), (2) when the teacher and students share the architecture (MLP -> MLP) and when the teacher has a different architecture (with better inductive biases for the task at hand) than the student (e.g. CNN -> MLP) [Note that we also have the MLP -> CNN setup]. The most interesting observations are when we compare (3) with both (1) and (2) and, especially when contrasting results from (3) with results from (2).  For instance, in (2),  we may have the knowledge of the teacher about inter-class relationships (which is modelled based on the inductive bias of the teacher) transferred to the student, but the question is that if in setup (3), where teacher and student are from completely different model classes, and the student doesn’t have the inductive bias of the teacher, can we still successfully transfer such a knowledge.  We update the paper to reflect this point and hope this makes the differences between our paper and previous works that study KD more clear.
> ***
> ## Response to comment #5
> ```>> 2. Note that the ECE results don't tell readers much. People expect soft labels to help not because they make models better calibrated, but because they boost performance, and it's not clear if people think better calibration leads to better main performance. Even if people do, in Fig 3(b) the ECE improves quite a lot for Transformer student with better teachers, so it is wrong to claim "Given the lack of significant improvement in ECE...".```
>
> Yes. It is correct that accuracy and calibration score are not necessarily correlated, and that is actually a good news for our purpose as they reflect different properties of a given solution.
> Regarding the sentence about Figure 3 that the reviewer refers to, i.e.,  “Given the lack of significant improvement in ECE in the self-distillation experiments (Figure 3b), it is more likely that the cause of the improvement in ECE when distilling LSTMs into Transformers is beyond the label smoothing effect of KD.”
> We think there is a misunderstanding as we state given the lack of significant improvement in ECE in the **self-distillation** experiments [...]. Self-distillation refers to the cases were the teacher and students are the same, e.g. Transformer -> Transformer, or LSTM -> LSTM. So we do not have “better teacher” in this case and we also observe no significant improvement in ECE, while when the teacher is a better model for the task (e.g. LSTM -> Transformer or UTransformer-Seq -> Transformer, etc.) , we see an improvement in the ECE. This is exactly what we also talk about in the previous comment, that KD with a better teacher leads to better ECE, while KD with a similar teacher does not.
> On a slightly relevant note, we also want to add that while performance is important, ECE can become a key metric when people want to deploy a model in real world application / production as most of the time, having a model with 80% accuracy, but with high confidence for the correct predictions and low confidence for the incorrect predictions is way better than a model that has 90% accuracy but it generates incorrect prediction with high confidence. This has been one of the motivations that we included calibration as a proxy of the quality of solutions different models converged to.

---

> > ### Comment · AnonReviewer2 · 2020-11-24
> > **Additional feedback on #4,5**
> >
> > # 4
> > I agree contrasting (2) with (3) may prove your point better, and the new version illustrates this point much better. The problem is, much like in comment #8, the teacher is simply better at the task, so I can just say "well maybe the inter-class correlation from the CNN is better modeled than that in the MLP, so although CNN->MLP is better than MLP->MLP, it could be just an effect of a better inter-class correlation". Now that #8 is addressed in your update, other experiments may benefit from a similar treatment (e.g. compare a Transformer and LSTM with same perplexity but different mu-accuracy as teachers). Note that this still may not be enough to convince all readers (e.g. reviewer 1 has the same major concern)
> >
> > # 5
> > Thanks, point taken.

---

> ### Author Response · Authors · 2020-11-18
> **Responses to comments #6, #7, #8 (from reviewer #2)**
>
> ## Response to comment #6
> ```>> The CNN/MLP experiment only has tasks that CNNs outperform MLPs. It would make it more interesting to see a task where the MLP outperforms CNN, e.g. a made up task whose ground truth is the xor of a few pixel positions, which could be hard for CNNs while easy for MLPs.```
>
> We agree that this would be very interesting [as a matter of fact, we got the same comment from one of the people who read our submission to provide feedback and we spent some time thinking about it]. However, it’s not easy to come up with such tasks. For our setup, we need a task on which both models have enough expressivity to learn that, but the inductive bias of one of them is more suited for the task. So for example, a task that only shows the difference between two models in terms of in-distribution test performance does not necessarily serve as a controlled setup for our analysis. If we understand correctly, in the XOR task, while the task might be harder for a simple CNN to learn because of the particular way of weight sharing in the model, it is not necessarily the case that MLPs, in particular, have the right inductive bias that allows them to generalize (to an out of distribution set) in this task.
>
> ***
> ## Response to comment #7
> ```>> Relatively small number of datasets. Just two datasets and two sets of networks is not very convincing to claim these findings generalize to other architectural changes.```
>
> Indeed it would always be nicer and more convincing to have more use cases. In this paper, given the limited space, we have chosen to go toward the direction of more in-depth analysis instead of extending the number of datasets explored. Furthermore, we show that our findings are consistent under various conditions (three different setups), for different input modalities, different model architectures, and different metrics. Given the limited space in the paper, we believe that the fact that our findings are consistent across all these different setups is more reassuring than having more experiments but with similar conditions.
> It is also noteworthy that in order to be able to make any conclusion from such experiments we need to have controlled setups where the effect of different factors could be studied in a semi-isolated setting. Both models and datasets should be carefully chosen so that the effects of the inductive biases of the models are clearly visible and justified. For example in a setup similar to the Corrupted-MNIST experiments, not only do we need both models to achieve relatively similar and well performance on the in-distribution data but also we need teacher models that achieve good out-of-distribution data, e.g. the CNN model that we use is carefully designed to be equivariant to translation and scaling. As to mention the difficulty of adding more datasets/setups: we would like to say that about a new dataset, using the simple CNN model we used, without applying any data augmentation (which adds noise to our analysis), it’s not possible to achieve proper performance even on a dataset as simple as CIFAR and also generalize to transformations such as translation and scaling. On the other hand, about new models, off-the-shelf models like a standard ResNet trained on the commonly used datasets do not show strong generalizability to out-of-distribution data.
>
> ***
> ## Response to comment #8
> ```>> Experiments are sometimes not apple-to-apple comparisons. And some experiments are not convincing or irrelevant.  The MNIST experiments only have two networks, and arguably CNN is absolutely better than MLP. It would make the point clearer if a worse CNN is used such that the MNIST-vanilla performance is the same as the MLP, and show improved generalization results on MNIST-C.```
>
> Although both MLP and CNN are rather well-performing models on the in-distribution MNIST [CNN performance = 0.992 and MLP performance = 0.985], we agree that CNN is still a better model with respect to its performance on vanilla MNIST. In a more general way, it is an interesting question to see how the quality of the teacher in the in-distribution set affects the quality of the student in the out-of-distribution set. To study that, we have added an experiment to the revised version of the paper, in Appendix F, where we distill CNNs with different qualities (at different stages of training) into MLPs and see how differently they improve the generalization of the student model to MNIST-C. Based on our experiments, we observed that in general, the accuracy of the MLP students follows the same trend as the accuracy of its teacher. In other words, a better CNN teacher results in a better MLP student. Also, given the results of an independently trained MLP from Table-5 in the paper, we see that the benefit of training an MLP via distillation for better generalization on in- and out-of-distribution sets only kicks in when we have a CNN teacher with a quality more than a certain threshold.

---

> > ### Comment · AnonReviewer2 · 2020-11-24
> > **Additional feedback for #6,7,8**
> >
> > # 6
> > 1. I don't think your experiments care if it's an out-of-distribution test set or not (cf. your NLP experiments used two criteria as a substitute for in-dist. vs out-of-dist. aspects). You can still draw conclusions on different aspects. For example, if you joint train the two tasks (digit classification and select-pixel xor) and distill both tasks, it could work.
> > 2. Also I think the xor idea might as well work for out-of-dist. samples. CNNs will look at nearby pixels' context, while NLP knows which pixel is which with laser-focus precision because each pixel location is unique for an NLP. So when you distort the image, NLPs may still outperform in generalization.
> > 3. Maybe the above turn out to be untrue, but you never know until you do the experiments :)
> >
> > # 7
> > 1. There's FashionMNIST. Is any other dataset that do subject-verb consistency?
> > 2. Use less distortion? Use translation only to avoid sampling blurriness? As long as CNNs outperform MLPs on that level of distortion, it doesn't affect the idea of your paper.
> >
> > # 8
> > Thank you for the extra experiment. It may improve its clarity if you compared the data-learned MLP and the MLP-teacher MLP-student performance on the right of Fig. 13b.

---

> ### Author Response · Authors · 2020-11-18
> **[–] Responses to comments #9, #10 (from reviewer #2)**
>
> ## Response to comment #9
> ```>> Figure 1 does not tell readers much, because latent representation can be both inductive bias and regular representation power, and we already know that KD can improve the student's representation power. Same for the third bullet point in page 2.```
>
> We would like to point out that in Figure 1, what we visualize is how a model evolves during training in terms of the way it models the relation between examples based on their representations [i.e. representational space, more details on this in Appendix B]. And it’s less about the quality, but more about the trajectory that the model follows. The comparison between the three given cases implies that given two models A and B with different architecture and with completely different paths for learning the representational space (either good or bad), we can see that the trajectory of B can become similar to A if we use A as the teacher for B in KD setup. This is particularly interesting, because B is not exposed to the intermediate state of A via KD, but learns to follow a similar trajectory, which indicates such information (which is not about the quality of representation, but about the way a model evolves in terms of learning them) is encoded in the knowledge transferred via KD.
>
> As we also mentioned in the previous comments, one important point in our experiments is contrasting the result from the KD setup when teacher and student have similar architectures with the KD setup where they have different architectures and the teacher model has a stronger inductive bias compared to the student. This comparison enables to distinguish the effect of “KD” from the effect of “KD with a teacher with the right/stronger inductive bias”. For example, if the representational similarity plots in Figure 7, distilling from CNN to MLP, drives the MLP students toward the CNNs cluster in the representational similarity space, whereas distilling MLPs into CNNs or MLPs into MLPs does not have this effect on the student models.
>
>  ***
> ## Response to comment #10
> ```>> Suspect of cherry-picking results from which loss (LM or classification) to show. Figures 2,4 are experiments using the LM loss, and Figures 3,5 are using the classification loss, without giving a clear explanation why.```
>
> Thanks for raising this question. We made the explanation of the language modeling vs. classification setups for the first case study more clear in the updated version of the paper.
> Here we explain this to address your concern. In the paper, we treat the LM and classification setups as two separate sets of experiments. That’s correct that for both of them we use the same dataset, but they are somewhat orthogonal. In both these setups, we first study the effect of the recurrent inductive bias (Figures 2, 3), and later we show how the solutions these models learn changes when they are trained through KD (Figures 2, 4, 5).
>
> It is noteworthy that some of the experiments that we can conduct in the classification setup are not applicable in the LM setup and vice versa. For example, the comparison between different Transformer architectures because we use the autoregressive language modeling objective which is not possible to apply on Transformers without future masking. Or the calibration experiments are specific to classification setup [Figures 3 and 5]. Moreover, the comparisons between the *perplexity* and accuracy is not applicable for the classification setup [Figures 2 and 4]. So there is no cherry-picking, and in the first case study, simply some of the experiments are not applicable in the alternative setup.

---

> > ### Comment · AnonReviewer2 · 2020-11-24
> > **Additional feedback for #9,10**
> >
> > # 9
> > It is true what you show is interesting. My problem with it is that it does not distinguish contributions from the *known* effects of KD (in this case, e.g. the inter-class correlations) and from the inductive bias distilling effect. I can make exactly the same argument with inter-class correlations: "KD tells the student how the class scores correlate. We show in the underlying feature space, which is directly one layer below the output, data follow the same trajectory as the teacher. This shows the effect of class correlation on the feature space."
> >
> > # 10
> > Thank you for the explanation. It would make it clearer if you point out that with LM loss you can only compare perplexity and A-accuracy, but with classification loss you can only compare ECE and mu-accuracy.
> >
> > However, it will make it convincing if you present the data in the same way, e.g. use Fig. 3's format to present Table 3, and have a version of Fig. 4 with ECE and mu-accuracy for the classification loss model.

---

> > > ### Author Response · Authors · 2020-11-24
> > > **Thank you for the additional feedback**
> > >
> > > Thank you so much for reading our replies carefully, and for following up on your comments and for additional suggestions and feedback.
> > >
> > > Regarding #1 and #2, we agree that there is room for improving the introduction and abstract. The last version was still updated in a deadline rush, but we make sure that we devote enough time to polish them by making them more specific, in particular by losing a little bit of high-level and general descriptions in page 1 and adding more specific explanations e.g. on the models we used as well as their characteristics and differences.  We will also address #10  and make the  LM setup vs. Classification setup more clear.
> > >
> > > Regarding the #6, #7, #8, thank you very much for your suggestions. We just wanted to quickly mention that we are actively working on this line of research and willing to add more datasets/tasks for the setup we have in this paper as well. Hopefully we will have more experiments with more datasets in the updated version of the paper.
> > >
> > > We also wanted to share our opinion on #4 and #9.
> > > In #9, about distinguishing our contributions from the known effects of KD, you have mentioned that  "maybe the inter-class correlation from the CNN is better modeled than that in the MLP, so although CNN->MLP is better than MLP->MLP, it could be just an effect of a better inter-class correlation". A few recent works have studied the different ways in which KD can help, for example [1] breaks down the effects of KD into three main factor: (1) benefits inherited from label smoothing,  (2) example re-weighting based on teacher's confidence on ground-truth,  and (3) prior knowledge of optimal output (logit) layer geometry (i.e., inter-class relationships). In our paper, we do not add an extra item to this list to show why KD helps, but rather show that the properties of the solution of the teacher can transfer to the student through all these factors. To just emphasize our contribution, the interesting phenomena is that even though the MLPs can’t learn such good solutions on their own, they can learn them when provided with the supervision signals from CNNs. The metaphorical way of looking at this would be, when a human student can’t figure out how to solve a math problem on their own but when looking at the solution to the problem, they can understand and learn the solution. We will try to make this point more clear in the updated version of the paper.
> > >
> > > In #4, we believe the argument that “maybe the inter-class correlation from the CNN is better modeled than that in the MLP, so although CNN->MLP is better than MLP->MLP, it could be just an effect of a better inter-class correlation”, is not contradictory to what we show. When training two models with different architectures on the same dataset, with the same training setup, isn’t the difference between “how they pick the inter-class correlation” the product of their different inductive biases? In other words, we think if CNNs learn a better inter-class correlation than MLPs, it’s because they have  a better inductive bias compared to MLPs. At the end of the day, “the choices made in the architecture of CNN", i.e., its receptive field, the sharing of parameters, etc., which leads to its ability to be translation invariance, also leads to learning a better solution.
> > >
> > > With all this said, we agree with you that it would be interesting if we have a setup that allows us to completely disentangle the effects of the inductive biases of the teachers from their ability in fitting  the training distribution.  Along this line, we have this example in our paper, where the teacher is not as good as the student  in terms of an aspect that shows how well the model fits the distribution of data with respect to the training objective, i.e., LSTMs are worse than transformers in terms of perplexity. After KD, we see that transformers also get worse in terms of perplexity, but their accuracy on the subject-verb agreement task increases.
> > >
> > > In general, we really really appreciate that you spent time on reading our paper and enjoyed learning about your thoughts and suggestions.
> > >
> > >
> > > [1] Tang et al.,  Understanding and Improving Knowledge Distillation, https://arxiv.org/abs/2002.03532.

---

> > > > ### Comment · AnonReviewer2 · 2020-11-25
> > > > **Last feedback**
> > > >
> > > > Thank you for the appreciation. Please consider taking advantage of these suggestions and address all the confusion that is happening to rework the presentation of the paper.
> > > >
> > > > > The metaphorical way of looking at this would be, when a human student can’t figure out how to solve a math problem on their own but when looking at the solution to the problem, they can understand and learn the solution.
> > > >
> > > > Yes, vanilla KD shows that too, by showing smaller network can't achieve high performance but can with the help of soft labes from a larger network. Although the effect is not as drastic as the MNIST experiment.
> > > >
> > > > > ... is not contradictory to what we show. When training two models with different architectures on the same dataset, with the same training setup, isn’t the difference between “how they pick the inter-class correlation” the product of their different inductive biases
> > > >
> > > > The point is not they are contradictory, it is that it does not prove things. Let
> > > >
> > > > - A="improved teaching of inter-class label correlation"
> > > > - B="effect of teaching of better inductive bias"
> > > > - C="improved performance in CNN->MLP vs MLP->MLP"
> > > > - "X=>Y" = "making X happen causes Y to happen"
> > > >
> > > > My point is "conclusions in prior work can tell us A=>C, so your paper showing B=>C is not that interesting precisely because we know A=>C, and B=>A seems inevitable". The paper would only make a difference if the amount that C happens cannot be fully explained by the amount A changes that is caused by B. Otherwise, if the effect on C is completely explained by the change in A, the contribution shrinks to "we note that since A=>C and we want C, we can make A happen by doing B" which is rather limited.
> > > >
> > > > I'd also note that in your reply you seem to conflate "inter-class correlation" with "sample-wise class affinity judged by a model (aka the soft label values, which describes the entirety of the controllable variables on the teacher's side)".

---

### Official Review · AnonReviewer1 · 2020-11-02
**Some interesting but insufficient experiments with a fundamentally flawed presentation**

**Rating:** 3
**Confidence:** 5

**Review:**

I previously reviewed a version of this paper and unfortunately
the primary issues with it have not been addressed sufficiently.
While some parts have changed,  I will draw on relevant portions
of my previous review where appropriate.

This paper sets out to investigate the respective "inductive biases"
of LSTM and Transformer neural networks, two dominant model families
that are frequently employed in applied NLP tasks.
They also seek to compare the "inductive biases" of CNNs and MLPs.

The air quotes are placed here because all generalization and thus
any claim concerning the generalization performance of a model
necessarily concern (whether explicitly or implicitly) inductive biases.

However, we do not typically need to invoke the term "inductive bias"
in every single sentence in a paper just to discuss the comparative suitability
of some models for some tasks and the comparatively poor performance.

There are times when it's beneficial not just to talk about comparative performance
of models but to talk rigorously about inductive biases. In many settings we can
formally characterize the bias of a hypothesis, e.g. through learning-theoretic
complexity measures. However, here the term is used excessively with
fuzzy claims made about some models having "stronger" or "weaker" inductive biases
without invoking any concrete measure of the expressivity of a hypothesis class.

So the flaws with this paper are two-fold. First, I do not believe that the contribution
is sufficiently interesting to warrant publication. Second, I do not believe that the
current exposition is suitable for publication.

Throughout the authors confuse what has actually been showing in prior works
for what has been speculatively claimed in prior works. For example, the authors
refer to the better performance of LSTMs vs Transformers on a set of agreement tasks
as an inductive bias for learning syntactic structures.

The authors show plots that simply depict performance but describe them
as characterizing the bias-variance tradeoff (absent any discussion of variance).
The authors have a lengthy discussion of calibration that does not make much sense
and parrots incorrect claims from previous papers such as the bizarre claim that label smoothing
calibrates classifiers (it's rather easy to see how label smoothing could lower ECE
for an otherwise overfit classifier but how in general it does not calibrate and can
even decalibrate classifier.
The idea that calibration magically falls out of knowledge distillation
[[[
previously this review had the clause:
"or that any of these models is "perfectly calibrated" (a claim they actually make)"
however, as the authors rightly point out, this was a mistake in my review,
the context here was defining a marginally calibrated classifier,
not in claiming that the KD models achieved perfect calibration.
]]]
is bizarre and unacceptable in a proper publication.

The knowlegde distillation experiments are interesting but the speculative interpretations
go far beyond what the actual experiments show (that distilling from a better performing teacher model
gives a better performing student model, regardless of the student's architecture).

In short, this paper is not suitable for publication and must be substantially rewritten.
The authors need both a more compelling result and a more forceful editor.
The authors train {LSTMs, Transformers, etc} for some tasks
 where they have been shown to perform better,
and show that downstream student models benefit from better teachers,
but this doesn't substantiate what strikes me as a strained framing of the material

======================================================================

UPDATE AFTER READING THE REBUTTAL

======================================================================

The rebuttal was thoughtful and detailed, and caught one careless error in my review
and I appreciate the author's care. At the same time the rebuttal itself contained
many conceptual flaws and failed to alleviate many core concerns.
Nevertheless, I think it warrants a minor increase in my score from 2 to 3.

---

> ### Author Response · Authors · 2020-11-18
> **Response to reviewer #1**
>
> We thank reviewer #1 for the time they spent reviewing our paper. We read the reviews precisely and prepared responses for each point that was raised by the reviewer. We will reply to each point in a separate comment to make it possible to discuss each point separately, with the hope that the conversation will lead to a fair conclusion.
>
> ## Response to comment #1
> ```>> I previously reviewed a version of this paper and unfortunately the primary issues with it have not been addressed in the slightest. While some parts have changed, I will draw on relevant portions of my previous review where appropriate. This paper sets out to investigate the respective "inductive biases" of LSTM and Transformer neural networks, two dominant model families that are frequently employed in applied NLP tasks. They also seek to compare the "inductive biases" of CNNs and MLPs.```
>
> We would like to emphasize that the main goal of our paper, as it’s quoted in the intro is to empirically investigate the answer to this question: “In Knowledge Distillation, are the preferences of the teacher that are rooted in its inductive biases, also reflected in its dark knowledge [Dark knowledge refers to the information encoded in the output logits of a neural network (Hinton et al., 2015)] , and can they thus be transferred to the student?”.
> 1. As the first step, based on ML literature, we set up scenarios in which inductive biases of models are known to be the key for success and failure of different models. Here are the two scenarios:
> LSTM vs. Transformer in the subject verb agreement:
> The “subject verb agreement” task has been presented in [2] as a “syntax-sensitive task” and used as a proxy assessing the ability of different models for capturing the syntactic hierarchical structure. Later,  in [2] and [3] LSTMs were compared to Transformers  against this task and the “recurrent inductive bias” were identified as the reason LSTMs generalize better in this task.
> [1] Assessing the ability of lstms to learn syntax sensitive dependencies, https://www.aclweb.org/anthology/Q16-1037/.
> [2] Tran et. al, The Importance of Being Recurrent for Modeling Hierarchical Structure, https://www.aclweb.org/anthology/D18-1503.
> [3] Dehghani et. al, Universal Transformers, https://openreview.net/pdf?id=HyzdRiR9Y7.
>
> 2. CNN vs.  MLPs  when evaluated on an out-of-distribution set. The models need to be invariant to transformations such as translation and scaling to generalize to these OOD test sets.
> The most well-known inductive bias of CNNs is their equivariance to translation [4] due to the particular form of parameter sharing, and given this property of this class of models, they can generalize to test sets where an arbitrary translation transformation is applied on examples without being exposed  to the translation transformation in the examples during training. However, MLPs fail to do so.
> [4] Goodflow et. al, Deep Learning. https://www.deeplearningbook.org/.
>
> So we would like to point out that, “ investigating the respective "inductive biases" of LSTM and Transformer / comparing the "inductive biases" of CNNs and MLPs” has been done before and we do not present or claim these as main new findings in our paper.  Although, we have done this a bit more in depth, for instance report mean and variance of accuracy, calibration, and other metrics like perplexity. Furthermore, for the scenario that involves the recurrent inductive bias, we study that in a more precise setup than it has been investigated before,  by identifying the different sources of this bias and empirically showing the role of each (Section 2), which is one of the contributions of our paper.
>
> Thus, what is described by the reviewer is only step zero to show that the claims from previous works hold  in our case studies  (in Section 2.1 and Section 3.1) and this lays the ground for the next set of steps containing experiments and analysis that are around our main goal, which is answering the main question of the paper, as mentioned above.  We updated the introduction of the paper in the revised version to make this point more clear and better spell out the contributions and findings.

---

> > ### Comment · AnonReviewer1 · 2020-11-24
> > **Thanks**
> >
> > I would like to thank the authors for the time that they put into attempting to address many of the major points of concern in my review. I believe they are writing in good faith and appreciate the effort that they put in. However, the rebuttal has ultimately only reinforced my belief that this is not a paper than can be salvaged in rebuttal but rather one that requires a top-to-bottom rewrite. I encourage the authors to be more disciplined in their thinking and writing and to resolve many elementary points of confusion.

---

> ### Author Response · Authors · 2020-11-18
> **Response to comment #2 (from reviewer #1)**
>
> ```>> The air quotes are placed here because all generalization and thus any claim concerning the generalization performance of a model necessarily concern (whether explicitly or implicitly) inductive biases. However, we do not typically need to invoke the term "inductive bias" in every single sentence in a paper just to discuss the comparative suitability of some models for some tasks and the comparatively poor performance. There are times when it's beneficial not just to talk about comparative performance of models but to talk rigorously about inductive biases.```
>
> It has been a long standing discussion that inductive biases of models are the key to their generalization ability and  different models show different generalization behaviour due to their different  inductive biases . From  “The Need for Biases in Learning Generalizations”  by Tom Mitchell: “If totally unbiased generalization systems are incapable of making the inductive leap to characterize the new instances, then the power of a generalization system follows directly from its biases – from decisions based on criteria other than consistency with the training instances.”
>
> In our paper, we emphasize that the final performance is not the only indicator of  “the behaviour of models in terms of the solution they converged to”,  and thus we backed up our findings from multiple axes besides comparing the performance. Here are different aspects and metrics we study:
> 1. In case study #1, in the LM setup, we compare the perplexity vs. accuracy of models and show how different models have different behaviours with respect to this. Note that  perplexity and accuracy reflect different properties and for instance in our paper, we have two models one with better accuracy and one with better perplexity.  [Section 2.2]
> 2. In case study #1 (classification setup) and #2, we compare the calibration of different methods. We compute the ECE for different models in different setups.  Note that a model with good accuracy can have high ECE, and a highly calibrated model can have low accuracy. Thus, the quality of models in terms of ECE is orthogonal to their accuracy. [Section 2.2 and Section 3.2]
> 3. In case study #1 and #2, we use the representational similarity of the activations from the penultimate layers of different models as an indicator of the structure of their latent space and the way they model  the relations between different examples [Section 2.2 and Section 3.2] For case study #2, we also used this analysis to visualize the trajectory of models to their converged solution, by tracking this property during training [Section 3.2 - Figure 1].
> 4. In case study #2, we analysed the solution that different pairs of models [both with same or different architectures and both  when learned with KD or independently] converge to, and checked if they  belong to the same flat basin of the loss landscape or not. These analyses again are based on the properties of the solutions from different models, regardless of the accuracy of models. [Appendix C]
> 5. In case study #1, we compare the per example behaviour of different models, in terms of  error overlap on test examples. This can be also used to measure how different solutions are similar and this can be independent of their accuracy. For instance, for two given model A and B, the accuracy on the test set for both models can be exactly equal to 0.5, while the model A fails on the first half of the test set, and model B fails on the second half, which shows the solution they converged to are different. [Appendix E]
> 5. In case study #1 and #2,  we run each experiment multiple times, with different seeds, and beside the mean for accuracy, we report its variance, which is directly related to the fact that the stronger the inductive bias of a model is, the less sensitive it is to random variations like the initialization or the order of training examples.
>
> In each case study, all these different aspects and metrics are evaluated in all the possible setups for all the models [i.e. LSTM and Transformer as well as MLP and CNN]: (1) when trained independently, (2) when trained via KD, where the teacher has the same architecture as the student, and (3) when trained via KD, where the teacher has a different architecture than the student. Comparing the results from setup (1) and (2) with the setup (3), for each model, in each case study, we show that we can improve the performance as well as quality/properties of the solution when the teacher has the right inductive bias.
> We believe that a key strength of our paper is studying a wide range of quantitative and qualitative aspects of the solutions (6 items listed above + final accuracy) to look into our research question. Given these different aspects can be completely orthogonal (as we showed in our paper), it’s a bit unjust to count all these as “just talking about comparative performance”.

---

> > ### Comment · AnonReviewer2 · 2020-11-24
> > **Thoughts on overclaiming and vagueness**
> >
> > While I agree with reviewer 1 that the paper makes big claims about their findings generalizing into all sorts of inductive bias in general and very cryptic about what they mean by inductive bias, I think the issue is mainly constrained in the abstract and intro. In the experiments, the authors talks about specific aspects that the models generalize, which are specific instances of inductive bias which are more rigorous.

---

> ### Author Response · Authors · 2020-11-18
> **Response to comment #3 (from reviewer #1)**
>
> ```>> In many settings we can formally characterize the bias of a hypothesis, e.g. through learning-theoretic complexity measures. However, here the term is used excessively with fuzzy claims made about some models having "stronger" or "weaker" inductive biases without invoking any concrete measure of the expressivity of a hypothesis class.```
>
> We do try to explain the sources of the inductive biases that we investigate and our focus is to “empirically” show the effects of these inductive biases from different aspects.  Here we explain these biases again and show how one model has a stronger bias than the other one in each case study.
>
> *For CNNs and MLPs*:
>
> 1. At the beginning of Section 3,  we explain the inductive bias of CNNs in comparison to MLPs as it is known in the literature:
> “The particular form of parameter sharing in the convolution operation makes CNNs equivariant to translation (Goodfellow et al., 2016). Note that, we can view CNNs as MLPs with an infinitely strong prior over their weights, which says that first of all the weights for each hidden unit are identical to the weights of its neighbour with a shift in space, second, the weights out of the spatially continues receptive field assigned to each hidden unit are zero.”
> 2. We show that CNNs generalize better than MLPs to a set of OOD datasets to confirm their inductive bias toward solutions that are translation and scale invariant.
>
> 3. Regarding stronger vs. weaker inductive bias:
> a)  We run each experiment multiple times with different random seeds for parameter initializations and report the “variance” besides the mean for our metrics, i,e, accuracy and ECE, and, representational similarity (e.g. Table 5 and 6 and Figure 7). As discussed in our paper and also in previous literature [1, 2]  models with stronger inductive biases tend to converge to similar solutions when trained multiple times but with different initialization/random seeds. Lower variance in CNNs compared to MLPs can be an indicator of this fact.
> [1] McCoy et. al,  Does syntax need to grow on trees? sources of hierarchical inductive bias in sequence-to-sequence networks. https://arxiv.org/abs/2001.03632.
> [2] Dodge et. al, “Fine-tuning pretrained language models: Weight initializations, data orders, and early stopping”.  https://arxiv.org/abs/2002.06305.
> b) As another indicator of  *“stronger” vs. “weaker”* inductive bias we went beyond variance in the reported metrics,  and investigated the variance in the converged solution in terms of their flat basin in the loss surface. In Figure 8 in appendix C, we show that two MLPs when initialized similarly converge to different basins of attraction (due to other stochasticity  in the training process, like order of examples, dropout, etc.)  while two CNNs converge to the same basin of attraction. This variance in the solutions the models converge to is one of the indicators of the inductive biases of the models.
>
> *For LSTMs and Transformers*:
>
> 1. As one of the contributions of our paper, we identify the sources of the recurrent inductive bias.
>  (1) *Sequentiality* (2) *Memory bottleneck*, and (3) *Recursion*.
> While in the literature the term ``recurrent inductive bias`` is often used vaguely and without specific definition, we try to define it in terms of the restrictions that exist in the recurrent neural network architectures. Furthermore, we design experiments, to incorporate these restrictions in the Transformer architecture separately to examine the role of each of them on the quality of the solution the Transformers converged to (Figure 3).
> 2. Similar to the other case study, we also report the variance of metrics we report,  including accuracy, ECE, representational similarity. We show LSTM models have smaller variance compared to Transformers with respect to different metrics.

---

> > ### Comment · AnonReviewer1 · 2020-11-24
> > **Deeply flawed argument.**
> >
> > I believe the authors are fundamentally confused about the very definition of the bias-variance tradeoff.

---

> > > ### Author Response · Authors · 2020-11-24
> > > **Reply to the comment from reviewer #1**
> > >
> > > Thanks for reading our response and for your reply. We would appreciate it if the reviewer can elaborate a bit on parts of our argument/comments that are deeply flawed and which parts indicate the confusion that the reviewer refers to.

---

> ### Author Response · Authors · 2020-11-18
> **Response to comment #4 (from reviewer #1)**
>
> ```>>The authors show plots that simply depict performance but describe them as characterizing the bias-variance tradeoff (absent any discussion of variance). ```
>
> We agree  it’s not immediately clear how the plot in Figure 2 is relevant to how the  two different models have different bias-variance trade-offs. We cut the explanation short in the paper (due to space problem), but  Here, we explain this in more detail and we also updated the paper with this additional explanation to make this point clear.
> Before explaining the Figure 2, we would like to point out that throughout the paper, in different plots and tables, we can see that the solutions that different instances of LSTM models (with different random seed) converge to have less variance compared to Transformers,  with respect to all different performance metrics, i.e. accuracy, ECE and representation similarity.
> Related to this, Figure 2 presents an observation about LSTM vs. Transformer which is really interesting (besides what we use from this plot to show the effect of KD, later in Figure 4).  In Figure 2, for each model, we have two different settings for the hyperparameters of the model:  in one setting the models are small (fewer number of parameters) and in the second setting the models are big (more number of parameters). More parameters for a model (given fixed architecture) means richer hypothesis spaces.
> We observe that while for the LSTM, increasing the size of the model results in a better performance, for the Transformer increasing the number of parameters results in a worse performance.  This aligns with the bias-variance trade-off argument that when using a  model with weaker biases, if we fix (limit) the amount of data, richer hypothesis spaces may hurt the generalization because they increase variance. So here, increasing the parameters of the transformer, hurts its accuracy, while this is not the case for LSTMs. Of course it is not a theoretical argument but a more intuitive speculation of what could be the source of this difference in the behaviour of these models and how this can be related to the bias variance trade-off of these two different learners.

---

> > ### Comment · AnonReviewer1 · 2020-11-24
> > **Not clear.**
> >
> > I do not believe that this argument is coherent.

---

> > > ### Author Response · Authors · 2020-11-24
> > > **Reply to the comment from reviewer #1**
> > >
> > > Thank you for reading our response. We would really appreciate it if the reviewer can elaborate which part of the argument is not coherent according to their opinion.

---

> > > > ### Comment · AnonReviewer1 · 2020-11-24
> > > > **Bias-Variance Tradeoff**
> > > >
> > > > There are many ways to create a distribution with respect to which you can compute variance. But the source of stochasticity matters. The canonical bias-variance tradeoff exists even if you fix the random seed because it concerns the _variance of parameter estimates with respect to the sample_. By your accounting linear regression would have 0 variance. I believe you are looking at variance in performance with respect to the seed while (mistakenly) invoking the terminology of statistical learning in undisciplined and misleading ways.

---

> > > > > ### Author Response · Authors · 2020-11-24
> > > > > **Reply to the comment about the relation between figure 2 and bias-variance tradeoff**
> > > > >
> > > > > Thank you very much for the explanation.
> > > > >
> > > > > We would like again to emphasize the connection of Figure2 with the bias-variance trade-off,  which was the original point raised by the reviewer. Our explanation of how the observation in this figure relates to the bias-variance trade-off does not involve anything about the “variance of performance” of models. Also note Figure2 is the only place we talk about bias variance trade-off. Here is again our explanation:
> > > > >
> > > > > > Figure 2, for each model, we have two different settings for the hyperparameters of the model: in one setting the models are small (fewer number of parameters) and in the second setting the models are big (more number of parameters). More parameters for a model (given fixed architecture) means richer hypothesis spaces.
> > > > > We observe that while for the LSTM, increasing the size of the model results in a better performance, for the Transformer increasing the number of parameters results in a worse performance. This aligns with the bias-variance trade-off argument that when using a model with weaker biases, if we fix (limit) the amount of data, richer hypothesis spaces may hurt the generalization because they increase variance. So here, increasing the parameters of the transformer, hurts its accuracy, while this is not the case for LSTMs. Of course it is not a theoretical argument but a more intuitive speculation of what could be the source of this difference in the behaviour of these models and how this can be related to the bias-variance trade-off of these two different learners.
> > > > >
> > > > > Given the famous bias-variance trade-off plot (e.g. https://djsaunde.files.wordpress.com/2017/07/bias-variance-tradeoff.png), when we move from a small model to a big model, in terms of model size, we move on the x-axis of this plot (i.e., we increase the model capacity/complexity) and for Transformers, variance becomes a source of generalization error and the performance drops, but for LSTM, because it has stronger bias, when we add capacity, we gain performance. In other words, for LSTM, adding more parameters we get closer to the optimal capacity, while for Transformers, adding more parameters we get away from the optimal capacity, which means these two models are sitting on a different spot in the bias-variance plot. We hope this makes it more clear.
> > > > >
> > > > > Note that the part of the explanation here and in the paper that refers to the “variance of performance” of models with different random seeds does not refer to bias-variance trade-off and only refers to the fact that “different instances of a model with stronger inductive biases are more likely to converge to similar solutions, compared to different instances of a model with weaker inductive biases”. As we also observed that variance of accuracy and ECE is higher for MLP and Transformer compared to CNN and LSTM. This is likely due to the presence of many local minima in the loss surface that are equally attractive to low-bias learners [1], also shown in some of the previous works we cited in our paper when we explain this.
> > > > >
> > > > > Although we believe this is not central to any of our findings if the reviewer finds it critical we are happy to also run additional experiments to report the variance of the models when the only source of stochasticity is in the data.
> > > > >
> > > > > [1] McCoy et al., https://arxiv.org/abs/1911.02969

---

> ### Author Response · Authors · 2020-11-18
> **Response to comment #5 (from reviewer #1)**
>
> ```>> The authors have a lengthy discussion of calibration that does not make much sense and parrots incorrect claims from previous papers such as the bizarre claim that label smoothing calibrates classifiers (it's rather easy to see how label smoothing could lower ECE for an otherwise overfit classifier but how in general it does not calibrate and can even decalibrate classifier.```
>
> Regarding the “the bizarre claim that label smoothing calibrates classifiers”, we would like to point out to a valuable previous work (which we referred to in our paper) by Müller, Kornblith,  and Hinton [1] where they carefully study the effect of label smoothing from different aspects and show that label smoothing can result in models with better calibration [From the introduction: “We demonstrate that label smoothing implicitly calibrates learned models so that the confidences of their predictions are more aligned with the accuracies of their predictions.”] .
> [1] Müller et. al, “When Does Label Smoothing Help?”, https://papers.nips.cc/paper/2019/file/f1748d6b0fd9d439f71450117eba2725-Paper.pdf
>
> We really appreciate it if the reviewer points us to a previous work or a theoretical discussion, or an empirical study that shows “label smoothing could lower ECE for an otherwise overfit classifier but how in general it does not calibrate and can even decalibrate classifiers”?
>
> More importantly, the fact that label smoothing helps calibration or not is not the center of the discussion about confidence calibration in our paper.  As a matter of fact, we are arguing that better calibration for student models is not merely the effect of label smoothing which happens in KD, and the better calibrated teacher (with more informative soft targets) leads to a better calibrated student model. While when the teacher is not well calibrated, although we still work with soft targets in KD (similar to label smoothing), we don’t see such an improvement in the calibration.

---

> > ### Comment · AnonReviewer1 · 2020-11-24
> > **Disagree**
> >
> > You cannot just assert by fiat that a mathematical claim is true. Calling the paper "valuable" does not make its assertions valid. Hinton and friends are just plain wrong here. Neural networks happen to be "overconfident" in practice, so label smoothing tends to improve measures like expected calibration error by making predictions less extreme. But this does not actually calibrate them and is clearly an incoherent procedure for achieving calibration.
> >
> > You asked for a theoretical discussion but an elementary exercise will suffice. Imagine a dataset with one dichotomous feature X that takes value 0 or 1, and the label takes value 0 when the feature is 0 and takes value 1 when the feature is 1. Now apply label smoothing, with say epsilon>0, say .2. The classifier will always output predictions with 80% confidence but will be accurate 100% of the time.
> >
> > Calibration is not something that can be achieved blindly by trial and error. Like "we did A, B, C, D, E, F ad hoc things and the resulting model happened as measured by say ECE to be slightly more calibrated".
> >
> > You are responsible as an author for making accurate claims, regardless of what slipped past peer review in the past.

---

> > > ### Author Response · Authors · 2020-11-24
> > > **Reply to the comment form reviewer #1 about the relation between label smoothing and calibration**
> > >
> > > We respectfully disagree with the reviewer on this point. We believe it is important to be able to build on top of the previous works, and if there is a criticism of a previous study, the most impactful way is to present it in the form of a scientific article so that future studies can refer to it. We also believe that empirical studies are as valuable as all the other types of works in machine learning, as some of the assumptions we make in our theoretical examples/discussions are much simpler than what happens in practice in real world setups (like when we assume a simple setup where a model has no error and perfectly learns the underlying distribution or when we assume there is no label noise).
> > >
> > > Regardless of that, we think explaining “whether label smoothing helps calibration” is out of the scope of our paper. We would like to again emphasize the context in which we cited previous works on this:
> > >
> > > In our paper, we have no experiment that shows label smoothing helps calibration. We have an experiment in which we showed that KD (using soft labels) with a more calibrated teacher leads to a more calibrated student. In that argument, we refer to this fact that, although it’s shown before that label smoothing can improve the model calibration, in our case, even if that is true,  “using soft labels” is not the only reason for the improvement in our case given that we have similar cases where we use KD (soft labels), but we get no improvement in terms of calibration (when the teacher is not better than a student in terms of calibration).
> > >
> > > So this is not a key point for validating any of our claims. If the reviewer points us to any reference that shows the label smoothing makes a model uncalibrated, we will cite that reference in our paper as a related discussion and the findings and conclusions of our paper remain unchanged.

---

> ### Author Response · Authors · 2020-11-18
> **Response to comment #6 (from reviewer #1)**
>
> ```>> The idea that calibration magically falls out of knowledge distillation or that any of these models is "perfectly calibrated" (a claim they actually make) is bizarre and unacceptable in a proper publication.```
>
> We look into the confidence calibration of the models as a factor to characterize the quality of the solution different models converge to.  Before going to the KD setup, in Figure 3, we first show models with better inductive biases to solve the task can also be better calibrated, besides their better accuracy.
>
> Next, we show evidence that training a model (e.g. student = Transformer) with a teacher (e.g. teacher = LSTM) that has the right inductive bias for the task at hand (e.g. subject-verb agreement task)  improves the calibration, compared to (1) training the model directly from data, i.e. no KD, (2) training the model using a teacher with similar architecture (e.g. teacher = Transformer). This has been shown and discussed in both case studies and the results in both cases back up that KD, with a good teacher (not any teacher), leads to better calibration. *We would like to ask the reviewer to elaborate what part of this dissection is not acceptable to their opinion?*
>
> Also, reading the paper, there is no claim in about a model with perfect calibration. We agree that such a claim is bizarre and unacceptable and *would really appreciate it if the reviewer points us to the section, experiments, or plots that talk about a model with perfect calibration?*
>
> p.s.
> In Figure 5, we show how a presumably well-calibrated model would look like in that plot (green bars) for the sake of visualization, which is a common thing to do for improving the readability of the plot and it is obvious that it doesn’t refer to any model.

---

> > ### Comment · AnonReviewer1 · 2020-11-24
> > **Please read your own paper carefully**
> >
> > "if we bin the confidence scores and compute the accuracy for each bin, the accuracies are !!!perfectly correlated!!! with the confidence values"

---

> > > ### Author Response · Authors · 2020-11-24
> > > **Reply to reviewer #1 about the "perfect calibration" claim**
> > >
> > > Thank you for pointing to the sentence that you believe is where we make this claim.
> > > However, this sentence is selected from the part of the paper that is explaining the concept of calibration. Here we bring the full paragraph for other readers:
> > >
> > > > As another indicator of the quality of the solutions that different models converged to in the classification setup, we look into their confidence calibration. Confidence calibration captures how well the likelihood (confidence) of the prediction of the model predicts its accuracy (Guo et al., 2017). For a well-calibrated model, if we bin the confidence scores and compute the accuracy for each bin, the accuracies are perfectly correlated with the confidence values. The Expected Calibration Error (ECE) is computed as the distance between the calibration curve of the model and the perfect calibration curve (DeGroot & Fienberg, 1983).
> > >
> > > We want to emphasize again that none of the models in the paper are claimed to be perfectly calibrated.

---

> > > > ### Comment · AnonReviewer1 · 2020-11-24
> > > > **Sorry!**
> > > >
> > > > Thanks for pointing this out, clearly I missed the context when making this note in my review and should have looked more carefully when searching for it just now. I am glad for the context and you are right here. I will update my review accordingly.

---

### Official Review · AnonReviewer4 · 2020-11-03
**I think that the issues are interesting, but of limited scope.**

**Rating:** 5
**Confidence:** 3

**Review:**

In this manuscript, the authors investigate the power of KD to enable benefiting from the advantages of different models at the same time. It first talks about inductive bias can be crucial in some tasks and scenarios, and further show that when a model has the right inductive bias, we can transfer its knowledge to a model that lacks the needed inductive bias and indicate that solutions that the student model learns are not only quantitatively but also qualitatively reflecting the inductive biases of the teacher model. The paper is well written, but not easy to follow. The efforts of this study may help better learn and find suitable models for some AI issues.

Here are my comments:
1. It seems that these experiments are well planned but without more detail, it was challenging to thoroughly evaluate the proposed work.
2.  How the large data sets will be integrated and analyzed, and what might come from the collective analyses was not described in detail. Some concern was expressed regarding whether the large amount of data generated would address the defined goals of the manuscript. More details regarding the integration and analysis of the experimental findings would help clarify this.
3. I also wonder that how the batch effects among different data and data types will be taken care of, as the integration of multiple data types and analysis is the key to this model.
4. I wonder why the manuscript chooses these models.

---

> ### Author Response · Authors · 2020-11-18
> **Response to reviewer #4**
>
> We would like to thank reviewer#4 for their comment on our submission. In the following, we reply to each comment separately.
> ***
> ## Response to comment #1
> ```>> 1. It seems that these experiments are well planned but without more detail, it was challenging to thoroughly evaluate the proposed work.```
>
> We really appreciate it, if the reviewer can point out what are the details that are missing? We have included the details of the hyperparameters of the models and the training processes in the appendix of the paper, and will gladly update the paper to include the important details if any is missing. [Note that for complete reproducibility,  the code for all the experiment and analyses is uploaded as supplementary materials]
> ***
> ## Response to comment #2
> ```>> 2. How the large data sets will be integrated and analyzed, and what might come from the collective analyses was not described in detail. Some concern was expressed regarding whether the large amount of data generated would address the defined goals of the manuscript. More details regarding the integration and analysis of the experimental findings would help clarify this.```
>
> We really appreciate it if the reviewer can be more specific in this comment and elaborate what part/section of the paper this comment refers to.
> ***
> ## Response to comment #3
> ```>> 3. I also wonder that how the batch effects among different data and data types will be taken care of, as the integration of multiple data types and analysis is the key to this model.```
>
> Unfortunately, we are missing the connection between this comment and the content of our paper. We really appreciate it if the reviewer elaborates what “different data types” refers to? Also no part of our paper includes anything related to “integration of multiple data types”.
> ***
> ## Response to comment #4
> ```>> 4. I wonder why the manuscript chooses these models.```
>
> Thanks for your question. At the beginning of each case study, we explained how the tasks/senario/setup for that case study has been selected and we also describe  what models are chosen and why.  In general, for both cases, we have chosen the architectures such that (1) they have enough expressive power to learn the tasks and (2) they have different inductive biases and different generalization behaviours.
> In case study #1, we have chosen LSTMs and Transformers. As we mention at the beginning of Section 2, LSTMs and Transformers are the basic building blocks of many state-of-the-art models for sequence modelling and natural language processing. Transformers are an expressive class of models that do extremely well on many tasks where the training data is adequate in quantity, Several studies, however, have shown that LSTMs can perform better than Transformers on tasks requiring sensitivity to (linguistic) structure, especially when the data is limited. These models are interesting as they have different inductive biases and the recurrent inductive bias of LSTMs is in particular a key feature for solving the task in case study#1.
> In case study #2, we have chosen MLPs vs. CNNs. As described at the beginning of Section 3,  MLPs are one of the simplest forms of neural networks that are used in many architectures and by themselves they are used to solve simple tasks (like MNISt classification). CNNs are the de facto choice for processing data with grid-like topology. Sparse connectivity and parameter sharing in CNNs make them an effective and statistically efficient architecture. The particular form of parameter sharing in the convolution operation makes CNNs equivariant to translation.  The inductive bias of CNNs is the key for their generalization to out of distribution scenario we have in case study #2, while MLPs lack such a characteristic.

---

### Author Response · Authors · 2020-11-18
**Thanks for all the comments and feedback**

We would like to thank all the reviewers for the time they spent reading and reviewing our submission and for their feedback. We update the paper, per reviewers' requests and suggestions, with the following revisions:
1. Update the abstract to include more details on how our work is different from previous works that study knowledge distillation.
2. Update the introduction to structure the contributions and spell out the findings in a more explicit way.
3. Add Appendix F that presents the result for an ablation study on the “Impact of the Quality of the Teacher”.  (Requested by reviewer#2)
4. Add Appendix G that presents the result for an ablation study on the “Impact of the Dataset used in the Distillation Step”.  (Requested by reviewer#3)
5. Add more explanation to some of the results, e.g. how Figure 2 relates to the bias-variance trade-off.
6. Some minor updates on some of the phrasings across the paper to improve the writing.

We also reply to the comments and questions raised by the reviewers in separate responses.

---

### Decision · Program_Chairs · 2021-01-07
**Final Decision**

**Decision:**

Reject

**Comment:**

This paper studies through empirical analysis an interesting problem: distilling the (strong) inductive bias of a teacher model to the student model (of weak inductive bias). The main claim/finding is that not only the "dark knowledge" in the logits can be transferred, but also the inductive bias (e.g. recurrence in RNN and translation invariance in CNN) can be transferred to make the student model stronger. This conclusion looks not very surprising but does contribute some new ideas to the fields of both deep learning and transfer learning.

The paper receives insightful but controversial reviews. Throughout reading the lengthy rebuttal and discussions, the AC, as a neutral referee of both sides, thought that while some expressions in the discussions seem a bit urgent and strained, both reviewers and authors managed to participate in the academic debate with a professional attitude that focus only on the technical issues. These discussions are very extensive and helpful for drawing a thorough understanding of this paper, and of the important research problem.

After the public interactions with the authors, a private discussion was performed between all reviewers and the AC, and among the four reviewers, one argued for rejection, one voted for rejection, one voted for acceptance, and one argued for acceptance. The AC believed that one of the reject votes lacks enough support in the comments and thus discarded it.
However, due to the wild disagreement between the reviewers, as well as between the reviewers and the authors, the AC read the paper carefully. The AC's main points are as follows:

- The research problem is interesting, and this paper appears to be the first work that studies the inductive bias transfer problem.

- The paper has made its endeavor to try to delve into this problem, through providing with extensive empirical results and analyses.

- The biggest weakness of this paper is the experimentation approach towards quantitatively studying the inductive bias: comparing the teacher and student models through the relational similarity between the penultimate-layer representation is simply not enough to justify that the inductive bias has been distilled/transferred.

Two reasons:
+ Due to the expressiveness of neural networks, it is not hard for the student to resemble the teacher's representations; in fact, this is a quite common result repeatedly used by researchers and practitioners, even when the student is only a smaller model. While the idea of distilling inductive bias is interesting, it simply cannot be sufficiently justified by the current experimentation design.
+ Inductive bias is something encoding our prior knowledge about the learning task and is often effective during the whole training procedure, which cannot be refreshed by the training data. However, albeit the distilled student model (transformer or MLP) resembles the representations of the teacher model (RNN or CNN), it is not certain whether the "distilled inductive bias" can linger in the student model if you further fine-tune the student model to downstream tasks. That is, it is highly possible that such "distilled inductive bias" of the student model will be refreshed by the future training data. In contrast, if we directly fine-tune the teacher model to downstream tasks, their inductive bias (recurrence of RNN or translation invariance of CNN) will be retained successfully in the fine-tuned model.
Basically, through this thinking, it is not clear whether the inductive bias has be distilled. If the distilled thing is refreshed out, it is probably not the inductive bias. More experimentation or a formal quantification of inductive bias is highly necessary here.

While Reviewer #1 was a bit skeptical in the comments and discussions (regarding which I had a private discussion with him/her), some of his/her comments are reasonable and should be well addressed before this paper could be accepted:
- Be rigorous in scientific writing. While the experiments with bias-variance tradeoff and calibration are interesting and relevant, the key concepts were used with less care. It is good to expand the authors' understanding of these concepts to make sure what they actually refer to.
- Try to provide sufficient elaboration when you try to claim something. It is true that for now, in our field, there are quite a few papers claiming something very big in the title or abstract, but simply cannot fulfill their story through rigorous or sound technical study. I suggest to tone down some of the key claims such that "inductive bias can be transferred" if they are not clearly provable.

Finally, AC believes this paper studies a very interesting problem that may draw wide attention, and the paper is acceptable in a future version if the above comments are well addressed. Since this is already a resubmission (as mentioned by Reviewer #1), I'd encourage the authors to focus on the technical parts of the comments and revise the paper substantially before submitting to yet another top venue.